# Adaptive Social Learning via Mode Policy Optimization for Language Agents

**Minzheng Wang**[1,2]    **Yongbin Li**[3]    **Haobo Wang**[4]    **Xinghua Zhang**[3*]
**Nan Xu**[2]    **Bingli Wu**[3]    **Fei Huang**[3]    **Haiyang Yu**[3]    **Wenji Mao**[2,1*]

[1] School of Artificial Intelligence, University of Chinese Academy of Sciences
[2] MAIS, Institute of Automation, Chinese Academy of Sciences
[3] Tongyi Lab, Alibaba Group   [4] Peking University
{wangminzheng2023, wenji.mao}@ia.ac.cn,
{shuide.lyb, zhangxinghua.zxh}@alibaba-inc.com

## Abstract

Effective social intelligence simulation requires language agents to dynamically adjust reasoning depth, a capability notably absent in current studies. Existing methods either lack explicit reasoning or employ lengthy Chain-of-Thought reasoning uniformly across all scenarios, resulting in excessive token usage and inflexible social behaviors in tasks such as negotiation or collaboration. To address this, we propose an **A**daptive **S**ocial **L**earning (**ASL**) framework in this paper, aiming to improve the adaptive reasoning ability of language agents in dynamic social interactions. To this end, we first identify the hierarchical reasoning modes under such context, ranging from intuitive response to deep deliberation based on the cognitive control theory. We then develop the **A**daptive **M**ode **P**olicy **O**ptimization (**AMPO**) algorithm to learn the context-aware mode adaptation and reasoning. Our framework advances existing research in three key aspects: (1) Multi-granular reasoning mode design, (2) Context-aware mode switching in rich social interaction, and (3) Token-efficient reasoning with depth adaptation. Extensive experiments on the benchmark social intelligence environment verify that ASL achieves 15.6% higher task performance than GPT-4o. Notably, our AMPO outperforms GRPO by 7.0% with 32.8% shorter thinking chains, demonstrating the advantages of our AMPO and the learned adaptive reasoning ability over GRPO's solution. Our code and data are available at `https://github.com/MozerWang/AMPO`.

## 1 Introduction

Large Language Models (LLMs) have demonstrated exceptional reasoning capabilities in static domains with well-defined rules and deterministic answers, such as mathematics, code, and logical reasoning (Yang et al., 2024; Liu et al., 2024; Anthropic, 2024; Comanici et al., 2025). However, there exists a notable gap between the reasoning capabilities required in these problems and those in open-ended social interaction, especially in scenarios involving conflicting interests and negotiations driven by agents' long-term goals. LLM-based agents offer new opportunities for modeling dynamic social contexts by simulating human behaviors (Zhou et al., 2024; Wang et al., 2024) and developing sophisticated reasoning capabilities. Yet, succeeding in such environments not only demands coherent alignment with agents' long-horizon objectives, but also rapid adaptation to evolving situations—capabilities with which current LLMs still struggle (Zhang et al., 2024; Liu et al., 2025a).

Recent research efforts on *social intelligence in language agents* have primarily focused on two pathways: (1) End-to-end goal-oriented training, which involves LLM post-training through supervised learning (Wang et al., 2024; Zhang et al., 2025a), and (2) External planning integration, augmenting agents with plug-and-play planning modules (Deng et al., 2024; Li et al., 2024a; Liu et al., 2025a). These methods predominantly focus on the fast-reasoning paradigm, which offers a response without sufficient thinking processes. However, in dynamic and complex social contexts, such responses often fail to capture subtle cues or anticipate long-term costs and benefits (see Figure 5). Evidence from

---
*Corresponding authors.

cognitive science indicates that humans often pause for deliberation in such situations (Evans, 1996; Krull & Dill, 1996), suggesting that social agents may also gain from similar reasoning processes. Although Long Chain-of-Thought (Long-CoT) has been proven effective in several domains (Chen et al., 2025), this reasoning paradigm has yet to be introduced to the social intelligence tasks above.

Although existing Large Reasoning Models (LRMs), such as OpenAI-o1 (Jaech et al., 2024) and DeepSeek-R1 (Guo et al., 2025), have demonstrated impressive capabilities with Long-CoT across various reasoning tasks (DeepMind, 2024; Team, 2024; OpenAI, 2025), most of them exhaust their reasoning regardless of the input complexity. This style of employing LLMs' reasoning uniformly is insufficient for handling the dynamics and richness in agent social interaction. For example, not all social interactions between language agents necessitate deep reasoning (Thorngate, 1976), exhaustive reasoning may degrade performance as a consequence of overthinking. Therefore, underlying social intelligence tasks, it is vital to empower LLM-based language agents with the reasoning ability that adapts to dynamic social environments. This highlights the need for social learning that occurs through interactive social experiences, to learn from complex social behavior and support the discovery of social intelligence (Yang et al., 2010; Gweon, 2024).

In this paper, we propose the **A**daptive **S**ocial **L**earning framework (**ASL**) to empower language agents with the capability for adaptive reasoning, enabling them to effectively respond in accordance with the dynamics of social interaction context. Specifically, we first develop reasoning modes inspired by hierarchical cognitive control theory (Koechlin & Summerfield, 2007; Badre, 2008), covering a spectrum from intuitive response, through shallow and strategic thinking, to deep deliberation. Next, we perform the injection of reasoning modes, which consists of behavioral cloning for cold-start and RL-based adaptive reasoning mode enhancement. For RL-based enhancement, we contrapuntally develop the **A**daptive **M**ode **P**olicy **O**ptimization (**AMPO**) algorithm, which incorporates the mode-level and sample-level information into advantage estimation to strengthen the context-aware reasoning mode switching. In terms of reward, we design three types of reward functions, including answer reward, format reward, and answer length reward, providing feedback for choosing the appropriate reasoning mode and answer. Finally, experimental results show that ASL achieves the SOTA performances in comparison with strong baselines.

The main contributions are summarized as follows: (1) We propose ASL, the first adaptive social learning framework for language agents, which consists of hierarchical reasoning modes and tailor-designed reinforcement learning to empower language agents with adaptive reasoning ability in rich social context. (2) We develop AMPO algorithm, which integrates mode-level and sample-level advantage estimation for dynamic mode switching, and improves flexible inference and token efficiency. (3) Extensive experiments demonstrate the significant improvements of ASL, with the performance gains up to 15.6% over GPT-4o. Additionally, compared to GRPO, AMPO shows 32.8% decrease in token utilization on average, accompanied by 7.0% performance gain.

## 2 ADAPTIVE SOCIAL LEARNING FRAMEWORK

To empower social language agents with adaptive reasoning in dynamic contexts, we introduce the ASL framework as shown in Figure 1. First, inspired by Hierarchical Cognitive Control Theory, we design specialized reasoning modes that structure the social agent's cognitive processes (§2.2). Next, we employ mode behavioral cloning to train the LLM to accurately follow predefined reasoning modes(§2.3). Finally, we propose adaptive mode policy optimization, which leverages reinforcement learning to enhance both adaptive mode selection and reasoning capabilities(§2.4).

### 2.1 TASK FORMULATION

The social intelligence task is modeled as a sequential dialogue interaction between two role-playing agents, $\mathcal{P}_1$ and $\mathcal{P}_2$, each with distinct profiles and private social goals. We formulate this interaction as a partially observable Markov decision process (POMDP), $\mathcal{M} = (\mathcal{S}, \mathcal{A}, \mathcal{O}, \mathcal{T}, \mathcal{R})$. The state space $\mathcal{S}$ represents the social context, and the action space $\mathcal{A}$ consists of open-ended natural language utterances. At state $s_t$, the current agent $\mathcal{P}_i$ receives an observation $o_t \in \mathcal{O}$ containing the dialogue history and its agent-specific private information. Based on this, it samples an action (utterance) $a_t^i$ from its policy $\pi_i(\cdot \mid o_t)$. The environment then transitions from state $s_t$ to $s_{t+1}$ via the transition function $\mathcal{T}$. An entire episode, forms a trajectory $\tau = \{o_1, a_1, \dots, o_T, a_T\}$, concluding at a terminal

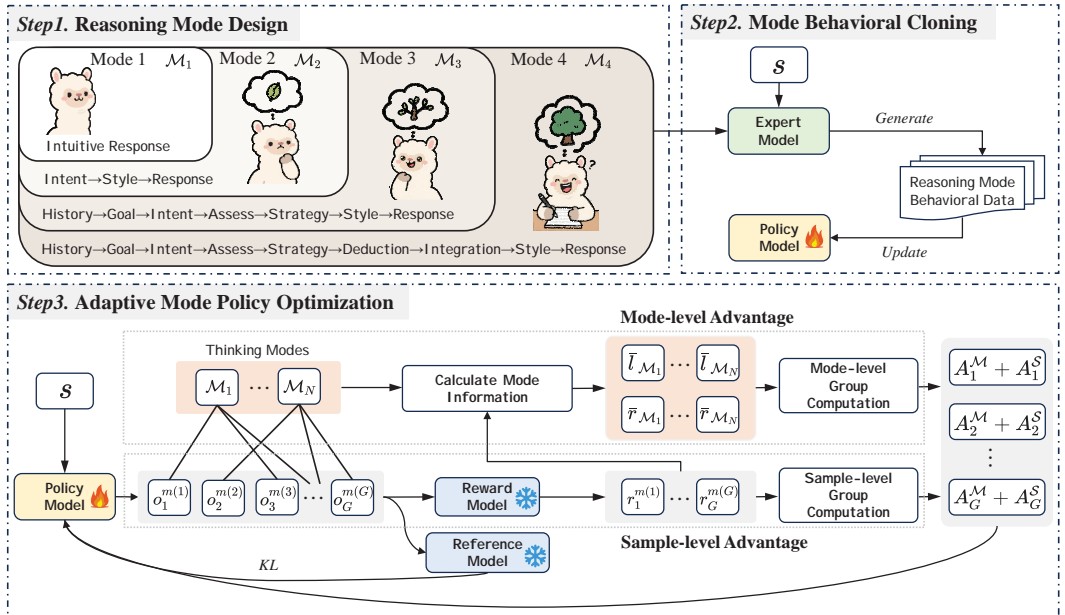

Figure 1: Demonstration of our **Adaptive Social Learning** framework, which consists of three steps: (1) **Reasoning Mode Design** based on Hierarchical Cognitive Control Theory, (2) **Mode Behavioral Cloning**, learning adherence to predefined reasoning modes and (3) **Adaptive Mode Policy Optimization**, introducing both mode- and sample-level advantage estimation during RL.

timestep $T$. At the end of the episode, each agent receives a terminal reward $R_i(s_T)$ from the reward function $\mathcal{R}$, which conducts a multi-dimensional evaluation on interaction quality with respect to each agent's private goals (e.g., building trust or resolving conflict). The objective for each agent is to learn a policy $\pi_i$ that maximizes its expected terminal reward: $\mathbb{E}_{\tau \sim \pi_i, \pi_{-i}}[R_i(s_T)]$.

## 2.2 REASONING MODES FOR LANGUAGE AGENTS

Existing SOTA LRMs exhibit significant limitations in social interaction (see our experimental results in Table 1). A detailed case study of the model outputs (see Appendix K.1) reveals several shortcomings in their reasoning trajectories, including unstructured thought processes, poor awareness of social goals, and an inability to controllably switch between different reasoning modes. To better equip LLMs for social scenarios, a more structured and sophisticated reasoning framework is necessary. We draw inspiration from Hierarchical Cognitive Control Theory (HCCT) (Koechlin & Summerfield, 2007; Badre, 2008), which provides a theoretical framework for understanding human cognitive behavior. HCCT posits that cognitive control operates through four distinct hierarchical levels, managing goals and actions at varying degrees of abstraction. Motivated by HCCT, we propose four corresponding levels of reasoning modes tailored for different social scenarios. As illustrated in Figure 6, these modes span from intuitive responses to progressively deeper levels of contemplation. The detailed mapping between our reasoning modes and HCCT's four hierarchical levels is provided in Appendix J. For each reasoning mode, we design specific and suitable actions aligned with linguistic principles:

**Mode 1 $\mathcal{M}_1$ (Intuitive Response)**: $\mathcal{M}_1$ is the most basic mode, characterized by intuitive responses based on learned associations and basic linguistic modes (Sacks et al., 1974; Norman & Shallice, 1985). It does not contain any reasoning actions, with only the final answer.

**Mode 2 $\mathcal{M}_2$ (Intentional Analysis)**: $\mathcal{M}_2$ is the basic interaction mode, focusing on understanding current intent and responding appropriately. $\mathcal{M}_2$ only requires maintaining basic interaction flow without complex strategic considerations. It encompasses a sequence of reasoning actions: ***Intent***, ***Style***, and ***Response***. *Intent* aims to analyze the other party's intentions (Grice, 1975). *Style* ensures consistency in the speaking style of the agent (Clark, 1996). *Response* gives the preliminary answer.

**Mode 3** $\mathcal{M}_3$ (**Strategic Adaptation**): $\mathcal{M}_3$ is the strategic reasoning mode, requiring speakers to not only understand immediate context but also comprehensively consider historical information, goal, and current situation assessment to formulate corresponding strategies. This enables speakers to better adapt to specific social situations. Compared with $\mathcal{M}_2$, $\mathcal{M}_3$ additionally introduces four reasoning actions: (1) **History** aims to analyze the history for better context understanding (Schiffrin, 1987). (2) **Goal** clarifies the agent's goal (Grosz & Sidner, 1986). (3) **Assess** analyzes goal alignment, round criticality, and improvement potential between parties (Brown, 1987). (4) **Strategy** enables the agent to propose a suitable strategy for the present social context (Clark & Brennan, 1991).

**Mode 4** $\mathcal{M}_4$ (**Prospective Deduction**): $\mathcal{M}_4$ is an advanced strategic simulation mode, requiring speakers to conceive multiple strategies and evaluate their effects through simulation, thereby making optimal decisions. $\mathcal{M}_4$ further introduces **Deduction** and **Integration** compared to $\mathcal{M}_3$. *Strategy* encourages the proposal of multiple strategies (Clark & Brennan, 1991), then simulating the execution of these strategies through *Deduction* action (Schank & Abelson, 2013). *Integration* action aggregates the results of *Deduction* for the preliminary answer. $\mathcal{M}_4$ facilitates the simulation of various situations to promote deeper thinking, effectively responding to more complex social contexts (Searle, 1969).

## 2.3 MODE BEHAVIORAL CLONING

To enhance the model's capability to adhere to four reasoning modes, we initially employ Behavioral Cloning (Bain & Sammut, 1999; Ross & Bagnell, 2010) to fine-tune the model as the foundation for subsequent reinforcement learning. Given a prompt corpus $\mathcal{X} = \{x_i\}_i^N$, we leverage an expert LLM to generate reasoning responses that are completely consistent with our reasoning modes and actions. To distinguish and control reasoning modes, each response begins with a special control token $c \in \mathcal{C} = \{<MODE\_1>, <MODE\_2>, <MODE\_3>, <MODE\_4>\}$. The LLM is thus to first generate the appropriate mode token and then produce a reasoning path consistent with that mode. The detailed data collection process and training parameters are described in Appendix G. Given the constructed dataset $\mathcal{D}_{bc}$, the objective is:

$$\mathcal{L}_{\text{BC}} = -\mathbb{E}_{(x,y) \sim \mathcal{D}_{bc}} \left[ \sum_{t=1}^{|y|} \log \pi_\theta(y_t \mid x, y_{1:t-1}) \right] \tag{1}$$

## 2.4 ADAPTIVE MODE POLICY OPTIMIZATION

GRPO (Shao et al., 2024) has proven highly effective for training of LRMs, which obviates the need for critic model and instead uses the average reward as the baseline to calculate the advantage:

$$A_{i,t} = \frac{r_i - \text{mean}(\{r_1, r_2, \ldots, r_G\})}{\text{std}(\{r_1, r_2, \ldots, r_G\})}. \tag{2}$$

However, this approach suffers from a critical limitation: it treats each sample independently, overlooking the inherent connections between samples in terms of their reasoning modes. For instance, a "step-by-step reasoning" response and a "direct response" are evaluated by GRPO solely based on their final reward $r_i$. Without additional mode-level information, this "mode-blindness" prevents the model from perceiving the trade-offs between different modes. Consequently, the model tends to converge towards fixed preferences rather than dynamically adjusting its reasoning modes according to specific scenarios. As shown in Figure 2a and Figure 2b, our experiments confirm that GRPO-trained models often resort to overly complex reasoning even for simple tasks.

To address this, we propose the AMPO algorithm. The core idea of AMPO is to incorporate both mode-level and sample-level advantage in its advantage estimation. The mode-level advantage guides the LLM in selecting the appropriate mode for a given scenario, while the sample-level advantage refines the reasoning trajectory generated within that chosen mode. This dual-level optimization enables the model to learn a dynamic and adaptive reasoning policy.

### 2.4.1 REWARD SHAPING

To provide a clear learning signal, the reward $r_i$ for each sample is carefully shaped and consists of three components: answer reward $r_i^a$, format reward $r_i^f$, and answer length reward $r_i^l$. The $r_i$ is

illustrated as follows. For detailed reward calculation and implementation, please refer to Appendix E.

$$r_i = \begin{cases} r_i^a \times r_i^l, & \text{if format is correct} \\ r_i^f, & \text{if format is incorrect} \end{cases} \tag{3}$$

**Answer Reward.** $r_i^a$ measures how well the answer improves the completion of the goal, assessed by an LLM evaluator. A boundary-aware scaling function is used to stabilize the reward signal to $[0, 1]$.

**Format Reward.** A large negative penalty ($r_i^f = -2$) is applied if the model's output deviates from the predefined structure of a reasoning mode. Otherwise, this component is neutral.

**Answer Length Reward.** In our early reward design, we observe that the LLM generates lengthy answers without achieving actual strategic improvements. Moreover, excessive answers lead to the accumulation of history in social interaction, significantly increasing computational costs. To solve this issue, we develop the $r_i^l$ to penalize answers that are longer than a target length, encouraging conciseness. A smooth penalty function maps the length deviation to a reward multiplier in $[0, 1]$.

### 2.4.2 ADVANTAGE ESTIMATION

**Mode-Level Advantage** $A^{\mathcal{M}}$ is designed to evaluate and select the optimal reasoning mode. An ideal reasoning mode should be both high-performing and efficient (Kahneman, 2011; Sui et al., 2025). Accordingly, the calculation of $A^{\mathcal{M}}$ embodies a trade-off mechanism:

$$A_{i,t}^{\mathcal{M}} = \begin{cases} \dfrac{\bar{r}^{m(i)} - \text{mean}(\{\bar{r}^{\mathcal{M}_1}, \bar{r}^{\mathcal{M}_2}, \ldots, \bar{r}^{\mathcal{M}_N}\})}{\text{std}(\{\bar{r}^{\mathcal{M}_1}, \bar{r}^{\mathcal{M}_2}, \ldots, \bar{r}^{\mathcal{M}_N}\})} & \text{if } \exists i,j \in [1,G] : r_i^{m(i)} \neq r_j^{m(j)} \\ -\tanh(\dfrac{\bar{l}^{m(i)} - \text{mean}(\{\bar{l}^{\mathcal{M}_1}, \bar{l}^{\mathcal{M}_2}, \ldots, \bar{l}^{\mathcal{M}_N}\})}{\text{std}(\{\bar{l}^{\mathcal{M}_1}, \bar{l}^{\mathcal{M}_2}, \ldots, \bar{l}^{\mathcal{M}_N}\})}) & \text{if } \forall i,j \in [1,G] : r_i^{m(i)} = r_j^{m(j)} \end{cases} \tag{4}$$

where $N$ denotes the total number of reasoning modes, and $G$ represents the total number of rollout samples, $m(i)$ represents the reasoning mode corresponding to $i$-th sample, $m(i) \in \mathcal{M} = \{\mathcal{M}_1, \mathcal{M}_2, ..., \mathcal{M}_N\}$. $\bar{r}^{\mathcal{M}_k}$ and $\bar{l}^{\mathcal{M}_k}$ are the average reward and average token length for all samples belonging to mode $\mathcal{M}_k$, respectively:

$$\bar{r}^{\mathcal{M}_k} = \frac{1}{|M_k|} \sum_{o_j^{m(j)} \in M_k} r_j^{m(j)}, \quad \bar{l}^{\mathcal{M}_k} = \frac{1}{|M_k|} \sum_{o_j^{m(j)} \in M_k} l_j^{m(j)} \tag{5}$$

where $M_k$ represents the rollout sample set of the $k$-th reasoning mode $\mathcal{M}_k$, $r_j^{m(j)}$ and $l_j^{m(j)}$ respectively denote the reward value and token length of the $j$-th sample. $o_j^{m(j)} \in \{o_1^{m(1)}, o_2^{m(2)}, .., o_G^{m(G)}\}$ where $\{o_1^{m(1)}, o_2^{m(2)}, .., o_G^{m(G)}\}$ is a group of outputs sampled from the old policy $\pi_{\theta_{\text{old}}}$.

The logic is as follows: (1) When average rewards ($\bar{r}$) differ across modes, the model is incentivized to choose the mode with higher performance. (2) When average rewards are similar (e.g., all modes successfully solve the task), the model is encouraged to prefer the more efficient mode—the one with a shorter average length $\bar{l}$. The *tanh* function is applied to the length-based advantage to normalize its value to the $[-1, 1]$ range, enhancing training stability.

**Sample-Level Advantage** $A^{\mathcal{S}}$ serves to refine generation quality within a given mode, defined as:

$$A_{i,t}^{\mathcal{S}} = \frac{r_i^{m(i)} - \text{mean}(\{r_1^{m(1)}, r_2^{m(2)}, \ldots, r_G^{m(G)}\})}{\text{std}(\{r_1^{m(1)}, r_2^{m(2)}, \ldots, r_G^{m(G)}\})}. \tag{6}$$

where $r_i^{m(i)}$ denotes the reward value of $i$-th sample $o_i^{m(i)}$ in the rollout group, $i \in [1, G]$. This component ensures that, for any chosen mode, the model is still driven to produce outputs that are better than the group average, thereby improving the quality of the chosen reasoning process.

### 2.5 POLICY OPTIMIZATION

With the dual-level advantage defined, the AMPO objective function integrates it into a PPO-style objective (Schulman et al., 2017; Shao et al., 2024). Our novel advantage directs the learning towards

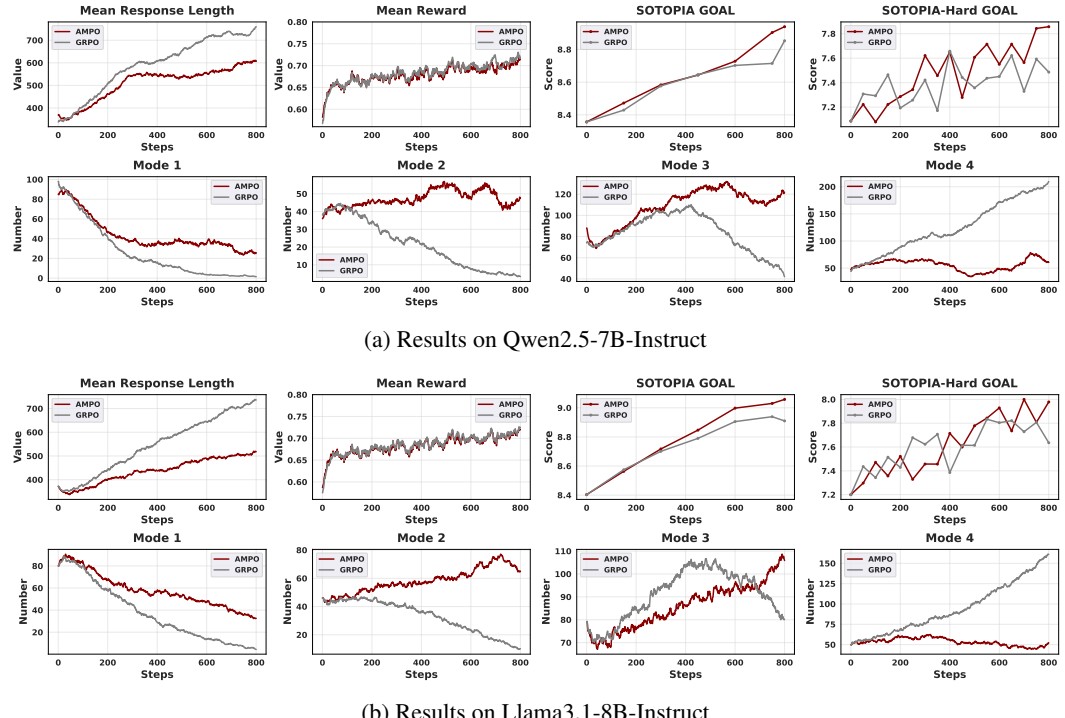

(a) Results on Qwen2.5-7B-Instruct

(b) Results on Llama3.1-8B-Instruct

Figure 2: Comparison of AMPO and GRPO on different base models in terms of training dynamics. (a) Results on Qwen2.5-7B-Instruct. (b) Results on Llama3.1-8B-Instruct.

both mode selection and content generation. The formal objective is:

$$\mathcal{J}_{\text{AMPO}}(\theta) = \mathbb{E}_{s \sim S, \{o_i^{m(i)}\}_{i=1}^{G} \sim \pi_{\theta_{\text{old}}}(o|s), m(i) \in \mathcal{M}} \left\{ \frac{1}{G} \sum_{i=1}^{G} \frac{1}{|o_i^{m(i)}|} \sum_{t=1}^{|o_i^{m(i)}|} \left\{ \min \left[ r_{i,t}^{m(i)}(\theta) \right. \right. \right.$$

$$(A_{i,t}^{\mathcal{M}} + A_{i,t}^{\mathcal{S}}), \text{clip}\left(r_{i,t}^{m(i)}(\theta), 1 - \epsilon, 1 + \epsilon\right)(A_{i,t}^{\mathcal{M}} + A_{i,t}^{\mathcal{S}}) \right] - \beta \mathbb{D}_{\text{KL}}\left[\pi_\theta || \pi_{\text{ref}}\right] \right\}. \tag{7}$$

where $A_{i,t}^{\mathcal{M}}$ represents mode-level advantage and $A_{i,t}^{\mathcal{S}}$ denotes sample-level advantage. The $\epsilon$ and $\beta$ are hyper-parameters, the ratio $r_{i,t}^{m(i)}(\theta) = \frac{\pi_\theta(o_{i,t}^{m(i)}|s, o_{i,<t})}{\pi_{\theta_{\text{old}}}(o_{i,t}^{m(i)}|s, o_{i,<t})}$ represents the importance sampling ratio and the KL divergence is calculated with the following unbiased estimator:

$$\mathbb{D}_{\text{KL}}\left[\pi_\theta || \pi_{\text{ref}}\right] = \frac{\pi_{\text{ref}}(o_{i,t}^{m(i)}|s, o_{i,<t})}{\pi_\theta(o_{i,t}^{m(i)}|s, o_{i,<t})} - \log \frac{\pi_{\text{ref}}(o_{i,t}^{m(i)}|s, o_{i,<t})}{\pi_\theta(o_{i,t}^{m(i)}|s, o_{i,<t})} - 1 \tag{8}$$

For computational efficiency, our online policy optimization adopts a single-turn training paradigm. Given a diverse corpus of social state $\mathcal{S} = \{s_1, s_2, ..., s_n\}$ where $s_n$ represents the current social context, including dialogue history and private information, the policy $\pi_\theta$ generates $G$ samples $o \sim \pi_\theta(\cdot|s)$ for each social context. The generated response receives a reward $r$ based on our reward shaping function, and the policy parameters are updated to maximize the AMPO objective in Equation (7). Detailed data preparation and the training process are provided in Appendix G.

## 3 EXPERIMENTS

### 3.1 EXPERIMENTAL SETTINGS

**Baselines.** Our method, implemented on both Qwen and Llama backbones, is evaluated against several baselines: (1) **Proprietary LLMs**, including GPT-4o (Hurst et al., 2024), Claude-3.5-Sonnet (Anthropic, 2024), and DeepSeeK-V3 (Liu et al., 2024); (2) **Large Reasoning Models**,

Table 1: Main results, showing AMPO's superiority over baselines. The highest score is highlighted in **bold**. The reported results are averaged over four runs (statistically significant with $p < 0.05$).

| Models | Self-Play | | | | GPT-4o-as-Partner | | | |
|---|---|---|---|---|---|---|---|---|
| | SOTOPIA | | SOTOPIA-Hard | | SOTOPIA | | SOTOPIA-Hard | |
| | GOAL ↑ | OVERALL ↑ | GOAL ↑ | OVERALL ↑ | GOAL ↑ | OVERALL ↑ | GOAL ↑ | OVERALL ↑ |
| *Proprietary LLMs* | | | | | | | | |
| GPT-4o | 8.19 | 3.76 | 6.97 | 3.46 | 8.19 | 3.76 | 6.97 | 3.46 |
| Claude-3.5-Sonnet | 8.29 | 3.71 | 6.33 | 3.09 | 8.42 | 3.77 | 6.64 | 3.30 |
| DeepSeek-V3 | 8.15 | 3.62 | 6.34 | 3.09 | 8.14 | 3.72 | 6.69 | 3.31 |
| *Large Reasoning Models* | | | | | | | | |
| OpenAI-o1 | 7.93 | 3.58 | 5.69 | 2.71 | 8.09 | 3.69 | 6.65 | 3.20 |
| OpenAI-o3-mini | 7.38 | 3.30 | 5.14 | 2.36 | 7.96 | 3.61 | 6.33 | 2.98 |
| Gemini-2.5-Pro | 7.85 | 3.43 | 5.67 | 2.55 | 8.12 | 3.59 | 6.70 | 3.09 |
| DeepSeek-R1 | 7.97 | 3.40 | 5.86 | 2.73 | 7.92 | 3.49 | 6.20 | 2.95 |
| QwQ-32B | 7.70 | 3.30 | 5.35 | 2.41 | 7.80 | 3.47 | 6.19 | 2.91 |
| Qwen2.5-7B-Instruct | 7.91 | 3.55 | 6.21 | 3.01 | 6.71 | 3.13 | 5.90 | 2.90 |
| w/ PPDPP (Deng et al., 2024) | 7.97 | 3.65 | 6.63 | 3.31 | 8.07 | 3.71 | 6.76 | 3.35 |
| w/ EPO (Liu et al., 2025a) | 8.09 | 3.51 | 6.82 | 3.12 | 8.41 | 3.86 | 6.81 | 3.51 |
| w/ DAT (Li et al., 2024a) | 7.97 | 3.59 | 6.39 | 3.10 | 8.11 | 3.70 | 6.78 | 3.36 |
| w/ DSI (Zhang et al., 2025a) | 8.35 | 3.75 | 7.31 | 3.51 | 8.15 | 3.70 | 6.87 | 3.42 |
| *Adaptive Social Learning* | | | | | | | | |
| w/ BC | 8.22 | 3.67 | 7.14 | 3.47 | 8.25 | 3.80 | 7.15 | 3.50 |
| w/ BC+GRPO | 8.87 | 3.85 | 7.44 | 3.41 | 8.52 | 3.92 | 7.20 | 3.50 |
| w/ BC+AMPO (Ours) | **8.95** | **3.87** | **7.85** | **3.54** | **8.60** | **3.94** | **7.50** | **3.65** |
| Llama3.1-8B-Instruct | 6.99 | 3.23 | 5.11 | 2.36 | 7.68 | 3.62 | 6.21 | 3.05 |
| w/ PPDPP (Deng et al., 2024) | 7.20 | 3.40 | 5.34 | 2.57 | 7.81 | 3.66 | 6.30 | 3.10 |
| w/ EPO (Liu et al., 2025a) | 7.94 | 3.78 | 6.79 | 3.27 | 8.37 | 3.83 | 6.91 | 3.53 |
| w/ DAT (Li et al., 2024a) | 7.34 | 3.44 | 5.89 | 2.85 | 7.78 | 3.65 | 6.18 | 3.03 |
| w/ DSI (Zhang et al., 2025a) | 8.08 | 3.64 | 6.93 | 3.31 | 8.08 | 3.67 | 6.84 | 3.41 |
| *Adaptive Social Learning* | | | | | | | | |
| w/ BC | 8.43 | 3.76 | 7.21 | 3.50 | 8.29 | 3.80 | 7.14 | 3.52 |
| w/ BC+GRPO | 8.86 | 3.84 | 7.59 | 3.44 | 8.63 | 3.92 | 7.30 | 3.54 |
| w/ BC+AMPO (Ours) | **9.08** | **3.95** | **8.06** | **3.68** | **8.75** | **3.98** | **7.68** | **3.74** |

including OpenAI-o1 (Jaech et al., 2024), OpenAI-o3-mini (OpenAI, 2025), DeepSeek-R1 (Guo et al., 2025), QwQ-32B (Team, 2024), and Gemini-2.5-Pro (DeepMind, 2024); (3) **Social Intelligence Methods**, including (a) PPDPP (Deng et al., 2024), which utilizes the policy planner to predict predefined strategies for reasoning; (b) EPO (Liu et al., 2025a), which employs the Strategic reasoning LLM to generate strategies in an open-ended action space; (c) DAT (Li et al., 2024a), which uses the trained planner to predict continuous vectors for controlling outputs; (d) DSI (Zhang et al., 2025a), which trains LLM through dynamic strategy injection learning; (4) **Variant of ASL Framework**, including (a) BC, behavioral cloning fine-tunes LLMs in our ASL framework and (b) GRPO, we use GRPO in the ASL framework for the RL (except for the advantage estimate, other settings are consistent with AMPO). For detailed baseline implementations, please refer to Appendix H.

**Benchmarks.** We evaluate our method on SOTOPIA and SOTOPIA-Hard (Zhou et al., 2024). SOTOPIA focuses on varying goal-oriented social interactions, while SOTOPIA-Hard challenges agents with complex strategic reasoning tasks. More detailed information is shown in Appendix F.

**Evaluation.** The social capabilities are evaluated across seven dimensions, among which the GOAL (ranging from 0 to 10) measuring how effectively a social agent achieves its goal. Following established research practices (Zhou et al., 2024; Wang et al., 2024; Liu et al., 2025a), we use GPT-4o as a proxy for human judgment to assess both **GOAL** and **OVERALL** performance (calculated as the mean of all seven dimensions), as studies have validated its high correlation with human evaluations. We set the temperature of the agents to 0.7 to encourage diversity of responses, and the temperature of the evaluator to 0 to ensure stable evaluation. We conduct evaluations under two settings: (1) **Self-Play**, where the social agent interacts with itself, and (2) **GPT-4o-as-Partner**, where the agent interacts with GPT-4o. For detailed evaluation settings, please refer to Appendix F.

## 3.2 EXPERIMENTAL RESULTS AND ANALYSIS

**RQ1: Is ASL framework effective for social agents?** As shown in Table 1, the proposed ASL framework achieves SOTA performance across all evaluated settings. For Llama backbone, AMPO even improves GOAL on SOTOPIA-Hard by 15.6% (6.97 → 8.06) over GPT-4o. The BC variant also delivers strong results, surpassing most baselines through supervised fine-tuning alone—highlighting

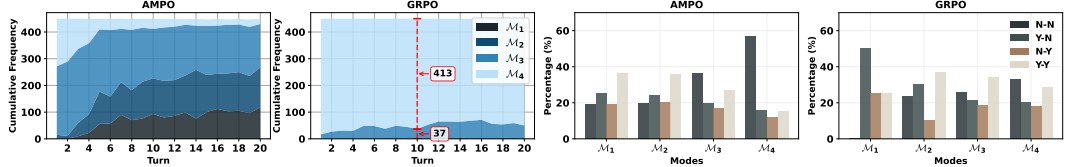

Figure 3: Analysis of adaptive behaviors. *Left*: distribution of modes across interaction turns. *Right*: distribution of modes across four context types — neither party succeeds (N-N), only our side succeeds (Y-N), only the other side succeeds (N-Y), and both parties succeed (Y-Y).

the effectiveness of our reasoning modes. Compared with Proprietary LLMs, the absence of explicit reasoning during social interaction limits response quality. LRMs, despite excelling in other domains, perform poorly in social scenarios. The case study (see Appendix K.1) reveals several limitations: poor awareness of social goals, insufficient history integration and circular reasoning patterns. By contrast, ASL's reasoning modes are explicitly aligned with social cognition patterns, which guides LLMs to produce appropriate reasoning trajectories. Concerning social intelligence methods, making strategy prompts alone proves insufficient to improve strategic execution. Overall, these findings underscore the ASL's superior capability in eliciting social reasoning for language agents, marking the first breakthrough about reasoning in the field of social intelligence.

**RQ2: Is AMPO more beneficial than GRPO for adaptive reasoning?** As shown in Table 1 and Table 2, AMPO exhibits significantly shorter responses than GRPO while achieving superior performance across all settings. Specifically, with Llama Backbone, AMPO's average inference length is 581 tokens—only 67.2% of GRPO's 865 tokens—yet it outperforms GRPO on the SOTOPIA-Hard benchmark by 7.0% ($3.44 \rightarrow 3.68$). As shown in Figure 2a and Figure 2b, during the training,

Table 2: Average tokens used per turn.

| Model | GOAL ↑ | Avg Tokens ↓ |
|---|---|---|
| QwQ-32B | 7.70 | 973 |
| DeepSeek-R1 | 7.07 | 711 |
| Qwen2.5-7B-GRPO | 8.87 | 905 |
| Qwen2.5-7B-AMPO | 8.95 | 647 |
| Llama3.1-8B-GRPO | 8.86 | 865 |
| Llama3.1-8B-AMPO | 9.08 | 581 |

while GRPO tends to converge to a single reasoning mode, manifested by a sharp increase in $\mathcal{M}_4$ and the eventual convergence of the other modes to zero, AMPO demonstrates awareness of dynamic context and adaptively explore diverse reasoning modes, effectively reducing the output length and achieving superior performance.

**RQ3: How do adaptive behaviors manifest in AMPO?** We conduct analysis of reasoning mode distributions from two perspectives—across interaction turns and across contexts—as shown in Figure 3. *(1) Mode distribution across turns.* Reasoning modes shift systematically over time: complex modes decline, while simpler modes rise. The most complex $\mathcal{M}_4$ is strongly front-loaded, appearing in 53% of turns 1–4 before dropping sharply. In contrast, $\mathcal{M}_1$ peaks late (50% in turns 14–20), and $\mathcal{M}_2$ remains frequent in mid-to-late turns (9–20). $\mathcal{M}_3$ distributes more evenly but decreases gradually from 31% in the first five turns to 21% in the last five. This pattern reflects task dynamics—complex reasoning dominates early when goals are unmet, while simpler reasoning suffices once goals are largely achieved. *(2) Mode distribution across contexts.* Simpler $\mathcal{M}_1$ and $\mathcal{M}_2$ occur mostly in straightforward contexts where both parties succeed, whereas complex $\mathcal{M}_3$ and $\mathcal{M}_4$ dominate in challenging contexts—particularly N-N, where neither party achieves the goal. This confirms AMPO's capacity to allocate reasoning depth according to scenario complexity.

**RQ4: How does ASL work?** To examine the effectiveness of the ASL design, we conduct a series of controlled variants, as reported in Table 3. *(1) Effect of answer length reward.* Removing the answer length reward ($r^l$) leads to much longer answers with decreased goal scores ($7.85 \rightarrow 7.46$), despite marginal gains in overall performance. This confirms that excessive verbosity is inefficient for social reasoning tasks, requiring significantly more tokens while failing to achieve better strategic outcomes. *(2) Effect of single reasoning mode.* Results of training with only a single mode reveal two trends: (i) Both performance and token usage increase as the mode depth grows from $\mathcal{M}_1$ to $\mathcal{M}_4$, with larger gains in challenging scenarios, indicating that deeper reasoning is beneficial for complex social contexts. (ii) While $\mathcal{M}_4$ achieves the better performance among single-mode settings, its token expenditure remains notably higher than AMPO with hybrid modes, underscoring the necessity of adaptive selection. *(3) Effectiveness of four hybrid reasoning modes.* Under GRPO, explicitly designed hybrid modes yield an 8.0% improvement in hard scenarios ($3.16 \rightarrow 3.41$) compared to mode-free reasoning, due to more structured and clearer reasoning guidance. AMPO further

Table 3: Comprehensive ablation study on ASL design components: effect of answer length reward ($r^l$), comparison of single reasoning mode ($\mathcal{M}_1$-$\mathcal{M}_4$), and effectiveness of hybrid reasoning modes.

| Qwen2.5-7B-Instruct | SOTOPIA | | SOTOPIA-Hard | | Avg Tokens ↓ |
|---|---|---|---|---|---|
| | GOAL ↑ | OVERALL ↑ | GOAL ↑ | OVERALL ↑ | |
| *Our Full Method* | | | | | |
| AMPO + w/ 4 Modes | 8.95 | 3.87 | 7.85 | 3.54 | 647 |
| *Effect of Answer Length Reward $r^l$* | | | | | |
| GRPO + w/o $r^l$ | 8.59 | 3.83 | 7.56 | 3.44 | 1705 |
| AMPO + w/o $r^l$ | 8.64 | 3.88 | 7.56 | 3.56 | 1617 |
| *Effect of Single Reasoning Mode* | | | | | |
| w/ Mode 1 $\mathcal{M}_1$ | 8.55 | 3.79 | 7.08 | 3.40 | 101 |
| w/ Mode 2 $\mathcal{M}_2$ | 8.71 | 3.42 | 7.28 | 2.80 | 572 |
| w/ Mode 3 $\mathcal{M}_3$ | 8.81 | 3.60 | 7.43 | 3.12 | 736 |
| w/ Mode 4 $\mathcal{M}_4$ | 8.86 | 3.80 | 7.62 | 3.31 | 972 |
| *Effectiveness of Four Hybrid Reasoning Modes* | | | | | |
| GRPO + w/o 4 Modes | 8.88 | 3.76 | 7.32 | 3.16 | 866 |
| GRPO + w/ 4 Modes | 8.87 | 3.85 | 7.44 | 3.41 | 905 |

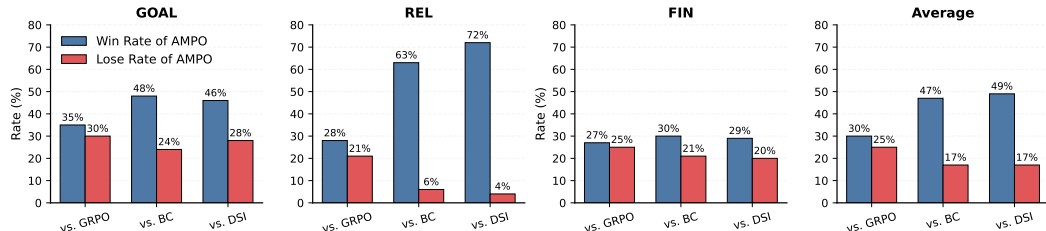

Figure 4: Human evaluation results. AMPO consistently outperforms GRPO, BC, and DSI across all dimensions (GOAL, REL, FIN, Average), with gains confirmed to be free of reward hacking.

boosts goal/overall scores by 5.5% (7.44 → 7.85) and 3.8% (3.41 → 3.54), while reducing token usage by 25%–29%. These results show that adaptive mode selection simultaneously delivers SOTA performance and substantial efficiency gains in dynamic contexts.

**RQ5: Human evaluation and case study.** To mitigate potential biases in LLM-based evaluation and assess possible reward hacking, we conduct rigorous human evaluations. We examine all episodes from both SOTOPIA and SOTOPIA-Hard, and ask three independent annotators to perform pairwise comparisons between AMPO and strong baselines (GRPO, BC, DSI) across three dimensions: Goal Completion (GOAL), Relationship (REL), and Financial/Material Benefits (FIN), with the average score reported. As demonstrated in Figure 4, AMPO consistently outperforms all baselines on every dimension. Our verification process (see Table 14 in Appendix I.2) confirms that these gains arise from legitimate strategy execution, with no evidence of reward hacking. We further conduct a case study (see Appendix K.2) illustrating how AMPO transforms Long-CoT reasoning into effective, goal-oriented social interaction. Consistent with quantitative results, AMPO advances dialogue objectives by fostering stronger interpersonal relationships and mutually beneficial outcomes—creating win–win situations that reflect superior strategic reasoning. Details of evaluation criteria and annotation guidelines are provided in Appendix I.

**RQ6: Additional Results.** We provide several supplementary analyses in the appendix to further validate the effectiveness of AMPO. First, we conduct out-of-distribution evaluations on the DEMO (Wang et al., 2025a) and NegotiationArena (Bianchi et al., 2024) benchmarks (see Table 4 and Appendix D.1). Second, we address potential concerns regarding long-horizon strategic consistency through single-turn optimization (see Table 6 and Appendix D.2). Third, we present detailed comparisons with efficient reasoning methods (see Table 7 and Appendix D.3). Fourth, we analyze the sensitivity of two critical hyperparameters: the length-reward coefficient and the target length (see Table 8 and Appendix D.4). Finally, we provide supplementary results using different LLM judges (see Table 9 and Appendix D.5).

## 4 RELATED WORK

**Social Intelligence**—the ability to pursue complex goals in dynamic interactions—is essential for human–AI collaboration (Bandura et al., 1986; Kihlstrom & Cantor, 2000; Tomasello, 2019; Sap et al., 2022). While LLMs are increasingly deployed as social agents capable of engaging in dynamic human interactions (Li et al., 2023; Ma et al., 2024; Xie et al., 2024; Li et al., 2024b), current static benchmarks (Sap et al., 2019; Zadeh et al., 2019; Shapira et al., 2023; Chen et al., 2024) fail to capture the nuanced, context-dependent nature of real-world social intelligence. SOTOPIA (Zhou et al., 2024) addresses this via a dynamic evaluation setting. Prior methods generally follow a fast-reasoning paradigm: (1) End-to-end goal-oriented training(e.g., SOTOPIA-$\pi$ (Wang et al., 2024), DSI (Zhang et al., 2025a)) improves social skills but lacks explicit strategy guidance, and (2) External planners (e.g., PPDPP (Deng et al., 2024), DAT (Li et al., 2024a), EPO (Liu et al., 2025a)) provide supervision but do not strengthen internal planning capabilities. These limitations motivate ASL, which enhances strategic execution via Long-CoT and adaptively switches modes for efficient social reasoning.

**Large Reasoning Models** Recent advances in LLMs have boosted reasoning through increased inference computation (Jaech et al., 2024; OpenAI, 2025) and RL post-training (Guo et al., 2025; Xie et al., 2025; Yu et al., 2025b; Chen et al., 2026), largely via Long-CoT reasoning (Wei et al., 2022; Ouyang et al., 2022; Muennighoff et al., 2025; Wang et al., 2025b; Aggarwal & Welleck, 2025; Wang et al., 2026). This paradigm achieves strong results in rule-based domains (Team, 2024; DeepMind, 2024; Zhang et al., 2025b; Tan et al., 2025b) but struggles in dynamic social environments (Thorngate, 1976; Liu et al., 2025a;b), often applying exhaustive reasoning regardless of task complexity (Sui et al., 2025) and not aligning with the selective nature of human decision-making in social contexts. While there is work addressing overthinking (Han et al., 2025; Luo et al., 2025; Yu et al., 2025a; Fang et al., 2025; Tan et al., 2025a), existing solutions remain limited to static domains such as mathematics, leaving a gap in dynamic, socially interactive settings. We present the first effective use of Long-CoT for social intelligence via our ASL framework, enabling context-aware mode switching for both effectiveness and efficiency.

## 5 CONCLUSION

This paper introduces the Adaptive Social Learning (ASL) framework, which represents the first effective realization of adaptive Long-CoT reasoning for social intelligence tasks. Drawing upon Hierarchical Cognitive Control Theory and linguistic principles, we establish four hierarchical reasoning modes. These modes encompass a spectrum of cognitive processes, ranging from intuitive response to deep contemplation. To enhance the context-aware mode switching and reasoning, we introduce the Adaptive Mode Policy Optimization (AMPO) algorithm, which integrates both mode- and sample-level information into advantage estimation. We conduct extensive experiments to demonstrate both the efficacy and distinctive advantages of ASL and AMPO. Furthermore, we validate the effectiveness of reasoning modes design and present a detailed analysis of AMPO's adaptive behaviors. To further validate our work, we employ rigorous human evaluation to provide additional verification of the effectiveness of our framework.

### ACKNOWLEDGMENTS

This work is supported in part by the National Natural Science Foundation of China under Grants #72293575, #72225011, #72434005 and #62206287, and Beijing Municipal Science & Technology Commission, Administrative Commission of Zhongguancun Science Park under Grant #Z231100007423016.

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

APPENDIX

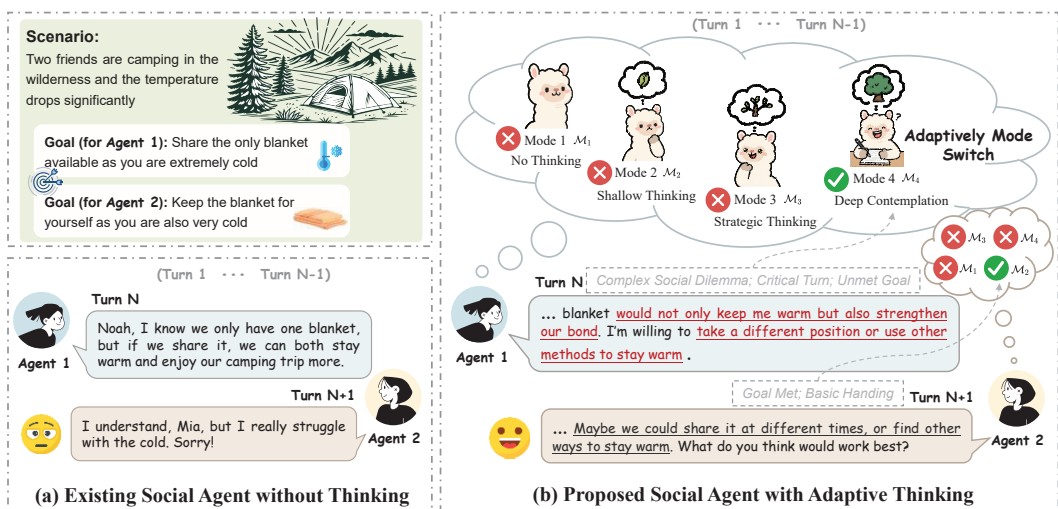

Figure 5: A social interaction example from SOTOPIA, comparing our method with existing approaches: (a) **Existing Thoughtless Social Agent** — relies on rapid, fast-reasoning inference without careful consideration of response strategies. This lack of strategic replies makes it difficult to achieve goals in situations involving conflicting interests. (b) **Proposed Thoughtful Social Agent** — employs adaptive reasoning, dynamically selecting the appropriate reasoning mode based on the current social context. By engaging in moderate reasoning before responding, our method is able to pursue social goals more effectively.

## A   THE USE OF LARGE LANGUAGE MODELS

During the preparation of this manuscript, the Proprietary LLM Claude-4.0-Sonnet is utilized to improve the linguistic quality of the text. Specifically, the model assisted with grammar checking, vocabulary adjustments, and improving the logical flow of expressions. All ideas, analyses, and conclusions presented in the paper are conceived and developed by the authors. The LLM's role is limited to language polishing and enhancing clarity of presentation.

## B   BROADER IMPACTS AND LIMITATIONS

Our proposed ASL framework presents a novel solution to address the challenges of adaptive reasoning in social intelligence domain, leveraging carefully designed reasoning modes and an innovative AMPO algorithm. The complete ASL training code and associated datasets will be made publicly available under the Apache-2.0 license. The code encompasses comprehensive training details for mode behavioral cloning and adaptive mode policy optimization, while the datasets include the BC and RL training data used in our experiments. These resources serve as valuable references and provide substantial support for researchers focusing on LLM-based social intelligence and extended Long-CoT reasoning capabilities.

In contrast to current social intelligence methods, ASL employs adaptive social learning to empower social agents with adaptive reasoning in dynamic social contexts, which achieves SOTA performance. However, reinforcement learning on LLMs still requires computational costs and hardware resources. Additionally, the inference time increases due to the need for generating more tokens, although we have reduced the total number of tokens compared to GRPO. These challenges are commonly encountered in test-time scaling methods.

Table 4: Results on Out-of-distribution benchmark. The Collab means the Collaboration set and the Non-Collaboration set from DEMO. AMPO yields consistent improvements over GRPO and the vanilla model in DEMO's multi-lingual dialogue element modeling tasks.

| Models | *Chat with GPT-4o* | | | *Chat with Claude-3.5-Sonnet* | | |
|---|---|---|---|---|---|---|
| | Collab ↑ | Non-Collab ↑ | Avg ↑ | Collab ↑ | Non-Collab ↑ | Avg ↑ |
| Qwen2.5-7B-Instruct | 7.73 | 7.62 | 7.65 | 7.57 | 7.62 | 7.61 |
| w/ GRPO | 7.90 | 7.70 | 7.72 | 7.81 | 7.84 | 7.83 |
| w/ AMPO (Ours) | **7.95** | **7.77** | **7.81** | **7.82** | **7.87** | **7.86** |
| Llama3.1-8B-Instruct | 8.56 | 7.68 | 7.92 | 7.89 | 7.54 | 7.63 |
| w/ GRPO | 8.35 | **7.96** | 8.06 | 8.09 | 7.81 | 7.89 |
| w/ AMPO (Ours) | **8.65** | 7.95 | **8.14** | **8.12** | **8.03** | **8.05** |

Table 5: OOD evaluation results on Sell&Buy and Ultimatum from NegotiationArena. AMPO yields consistent improvements over GRPO and the vanilla model on diverse negotiation tasks

| Model | Sell&Buy | | | Ultimatum | | |
|---|---|---|---|---|---|---|
| | Self Profit ↑ | Total Profit ↑ | Winning Rate ↑ | Self Profit ↑ | Total Profit ↑ | Winning Rate ↑ |
| Qwen2.5-7B-Instruct | 11.90 | 13.87 | 38.1% | 14.23 | 32.68 | 8.1% |
| w/ GRPO | 16.75 | 23.67 | 52.6% | 34.65 | 76.88 | 25.1% |
| w/ AMPO | **17.94** | **23.96** | **54.6%** | **35.71** | **79.23** | **28.4%** |
| Llama3.1-8B-Instruct | 3.12 | 3.82 | 9.4% | 14.43 | 22.41 | 15.9% |
| w/ GRPO | 14.99 | 23.58 | 48.7% | 31.64 | 69.68 | 18.4% |
| w/ AMPO | **15.57** | **24.00** | **50.2%** | **34.91** | **76.42** | **20.5%** |

## C  DATA STATISTICS

**Behavorior Cloning and RL Data** All training episodes are collected with SOTOPIA-$\pi$ (Wang et al., 2024) using scenarios disjoint from the test environment. SOTOPIA-$\pi$ contains 410 scenarios, which we split into 100 scenarios for BC and 310 for RL.

**Evaluation** The max interaction number is set to 20 turns. In the self-play setting, we run a single pass over all SOTOPIA tasks (450) and SOTOPIA-hard tasks (70). In the GPT-4o partner setting, we evaluate each task twice to balance speaking order between agents ($2 \times 450$ SOTOPIA and $2 \times 70$ hard tasks). For the DEMO benchmark (2,000 dialogue episodes), we likewise run each episode twice under each partner configuration (Chat with GPT-4o and Chat with Claude 3.5 Sonnet), yielding 4,000 runs per partner.

## D  ADDITIONAL EXPERIMENTS

### D.1  OUT-OF-DISTRIBUTION EVALUATION

To assess the generality of AMPO, we further evaluate on the *Dialogue Agent Interaction* task from the DEMO (Wang et al., 2025a) and Sell&Buy and Ultimatum task from the Negotiation-Arena (Bianchi et al., 2024) benchmark. The DEMO comprises 2,000 dialogue interaction episodes with a balanced 1:1 ratio of Chinese and English conversations, including 541 collaboration and 1,459 non-collaboration tasks. In contrast to SOTOPIA, DEMO emphasizes fine-grained dialogue element modeling, and contains more adversarial and non-cooperative scenarios, while explicitly incorporating Chinese language evaluation. We adopt the same evaluation protocol as defined in the original benchmark. For NegotiationArena, we employed a dialogue-based evaluation approach using GPT-4o as the interaction partner. Each task comprises 200 scenarios with role positions swapped between Agent1 and Agent2. To ensure result stability, each scenario was executed 4 times, yielding a total of 1,600 evaluations per task. Importantly, this setting constitutes an out-of-distribution evaluation, since none of the tested models were trained on DEMO and NegotiationArena. For fairness, we directly reuse AMPO and GRPO checkpoints previously trained and evaluated on SOTOPIA without any additional fine-tuning. We report goal achievement scores (maximum of 10) for DEMO, and Self Profit, Total Profit, and Winning Rate for NegotiationArena.

As shown in Table 4 and Table 5, across both collaboration and non-collaboration subsets of DEMO, LLMs trained with AMPO consistently outperform their GRPO and vanilla counterparts. Notably, these gains hold across both Chat with GPT-4o and Chat with Claude-3.5-Sonnet settings. Similarly,

Table 6: Ablation study on Our Design for Long-Horizon Consistency: effect of Diverse State and Goal-aware Reward.

| Qwen2.5-7B-Instruct | SOTOPIA | | SOTOPIA-Hard | | Avg Tokens ↓ |
|---|---|---|---|---|---|
| | GOAL ↑ | OVERALL ↑ | GOAL ↑ | OVERALL ↑ | |
| AMPO | 8.95 | 3.87 | 7.85 | 3.54 | 647 |
| AMPO + w/o Diverse State | 8.58 | 3.71 | 7.12 | 3.32 | 139 |
| AMPO + w/o Goal-aware Reward | 8.65 | 3.45 | 7.21 | 2.91 | 726 |

AMPO demonstrates stable performance improvements on NegotiationArena. These results provide strong evidence that the social intelligence capabilities acquired through AMPO transfer effectively to OOD dialogue element modeling and negotiation tasks.

## D.2 SINGLE-TURN OPTIMIZATION WITH LONG-HORIZON AWARENESS

To maintain computational efficiency, our policy optimization employs a single-turn training paradigm. We address potential concerns regarding long-horizon strategic consistency through three complementary perspectives: mechanistic analysis, paradigm validity, and empirical evidence.

**Mechanistic Analysis.** AMPO inherently captures long-term dependencies through deliberate design. At the data construction level, training data comprises dialogue states spanning diverse complexity tiers and temporal phases (detailed in Appendix G.2), with each sample containing a complete dialogue history, ensuring the model has sufficient contextual information during single-turn updates. At the reward designing level, the critical design lies in evaluating the marginal contribution of this turn toward overall goal completion (Equation (9)) rather than isolated response quality. Specifically, the boundary-aware scaling function dynamically adjusts reward magnitude based on current progress, ensuring effective learning signals across different stages; myopic responses are penalized for failing to advance ultimate goal completion, thereby embedding long-term objectives into single-turn optimization targets. Furthermore, the carefully designed reasoning modes within the ASL framework inherently guide prospective thinking: Mode 3 incorporates *Goal* and *Assess* actions to continuously evaluate goal alignment and turn criticality, while Mode 4 conducts multi-strategy prospective simulation through *Deduction* and *Integration*. These actions inject long-term planning capabilities into each single-turn decision. Our existing experiments on the Effect of Single Reasoning Mode (See Table 3) demonstrate that both Mode 3 and Mode 4 exhibit superior performance.

To validate the effectiveness of these designs, we conducted ablation experiments. The full results are shown in Table 6 When data diversity is removed (retaining only data after fixed turns), goal completion drops precipitously (SOTOPIA-Hard: 7.85 → 7.12), and output length degrades to 139 tokens, indicating that the model cannot perform effective reasoning or learn differentiated strategies across dialogue stages, thereby losing long-term planning capacity. When the marginal contribution reward function is removed (using only absolute scores), performance declines significantly (OVER-ALL: 3.54 → 2.91), while output length paradoxically increases to 726 tokens. This occurs because the model cannot perceive the marginal value of current actions toward final objectives, leading to suboptimal adaptation: it becomes overly cautious when close to goal completion and insufficiently strategic when far from target states, thereby losing dynamic responsiveness to varying dialogue states. These ablation experiments directly demonstrate the necessity of our design for achieving long-horizon consistency within single-turn optimization. Goal-aware Reward

**Paradigm Validity.** Decomposing multi-turn into single-turn optimization constitutes a well-established technical paradigm in the dialogue systems domain. The core advantage of this approach lies in achieving comprehensive coverage of the dialogue state space through turn-level exploration while significantly reducing training complexity to enable large-scale training. It is worth emphasizing that multiple seminal studies (Li et al., 2017; Su et al., 2019; Bao et al., 2020; Glaese et al., 2022; Su et al., 2022) have adopted similar multi-turn interaction decomposition with single-turn policy optimization training strategies and achieved superior performance.

**Empirical Evidence.** Our empirical results provide evidence that AMPO indeed exhibits long-horizon strategic consistency. First, dynamic mode adaptation demonstrates significant temporal awareness capabilities. As illustrated in Figure 3, AMPO exhibits a systematic pattern of "early deliberation—late maintenance" mode transitions. This mode evolution clearly indicates that the

Table 7: Comparison with efficient reasoning work. AMPO achieves the best performance with acceptable token usage across all evaluation settings.

| | Self-Play | | | | GPT-4o-as-Partner | | | | |
| | SOTOPIA | | SOTOPIA-Hard | | SOTOPIA | | SOTOPIA-Hard | | Tokens ↓ |
| Models | GOAL ↑ | OVERALL ↑ | GOAL ↑ | OVERALL ↑ | GOAL ↑ | OVERALL ↑ | GOAL ↑ | OVERALL ↑ | |
|---|---|---|---|---|---|---|---|---|---|
| Qwen2.5-7B-Instruct | 7.91 | 3.55 | 6.21 | 3.01 | 6.71 | 3.13 | 5.90 | 2.90 | - |
| w/ GRPO | 8.87 | 3.85 | 7.44 | 3.41 | 8.52 | 3.92 | 7.20 | 3.50 | 905 |
| w/ TALE-PT (Han et al., 2025) | 8.35 | 3.71 | 6.21 | 3.01 | 8.21 | 3.78 | 7.09 | 3.49 | 460 |
| w/ DeGRPO (Fang et al., 2025) | 8.46 | 3.75 | 7.04 | 3.37 | 8.47 | 3.86 | 7.16 | 3.48 | 1150 |
| w/ LS-Mix (Yu et al., 2025a) | 8.31 | 3.76 | 6.96 | 3.34 | 8.31 | 3.83 | 7.25 | 3.58 | 469 |
| w/ O1-Pruner (Luo et al., 2025) | 7.98 | 3.65 | 6.91 | 3.31 | 8.19 | 3.78 | 6.91 | 3.42 | 280 |
| w/ AMPO (Ours) | **8.95** | **3.87** | **7.85** | **3.54** | **8.60** | **3.94** | **7.50** | **3.65** | 647 |
| Llama3.1-8B-Instruct | 6.99 | 3.23 | 5.11 | 2.36 | 7.68 | 3.62 | 6.21 | 3.05 | - |
| w/ GRPO | 8.86 | 3.84 | 7.59 | 3.44 | 8.63 | 3.92 | 7.30 | 3.54 | 865 |
| w/ TALE-PT (Han et al., 2025) | 8.34 | 3.80 | 7.17 | 3.59 | 8.27 | 3.81 | 6.70 | 3.49 | 304 |
| w/ DeGRPO (Fang et al., 2025) | 8.71 | 3.92 | 7.64 | **3.72** | 8.55 | 3.90 | 7.53 | 3.70 | 1121 |
| w/ LS-Mix (Yu et al., 2025a) | 8.43 | 3.84 | 7.24 | 3.53 | 8.36 | 3.86 | 7.26 | 3.67 | 236 |
| w/ O1-Pruner (Luo et al., 2025) | 8.53 | 3.90 | 7.49 | 3.67 | 8.23 | 3.81 | 7.16 | 3.62 | 162 |
| w/ AMPO (Ours) | **9.08** | **3.95** | **8.06** | 3.68 | **8.75** | **3.98** | **7.68** | **3.74** | 581 |

model dynamically adjusts reasoning depth according to the overall dialogue progression. Second, performance validates strategic effectiveness. AMPO nearly achieves optimal performance across all experiments in this study, which strongly suggests that the model possesses long-term goal-oriented decision-making capabilities rather than myopic local optimization. Third, case analysis provides qualitative evidence. The 8-turn dialogue example in Section K.2 demonstrates that AMPO consistently centers on the core objective of helping a friend solve financial problems, with strategies progressing from emotional support (Turn 1) → resource provision (Turn 3) → action planning (Turns 5-7), presenting a clear progressive structure that embodies cross-turn strategic coherence.

## D.3 ADDITIONAL EFFICIENT REASONING BASELINE

To further assess the effectiveness of our approach, we conduct comparisons with a set of representative reasoning-efficiency methods originally proposed to mitigate overthinking. Existing works focus on static domains such as mathematical problem solving, where answers are fixed and outcome evaluation is straightforward. To the best of our knowledge, AMPO is the first method specifically tailored to optimize reasoning efficiency in social interaction tasks. The baselines include: (1) **O1-Pruner** (Luo et al., 2025), which uses reinforcement learning with a Length-Harmonizing Reward that combines length reduction incentives and accuracy penalties to fine-tune long-thought reasoning models for efficient inference.; (2) **TALE-PT** (Yu et al., 2025a), which is a post-training approach that internalizes token-budget awareness into LLMs, enabling them to generate more token-efficient reasoning responses without explicit budget constraints in prompts; (3) **LS-Mix** (Han et al., 2025), achieves efficient reasoning without overthinking via post-training on both long and structure-preserved short chain-of-thought data; (4) **DeGRPO** (Fang et al., 2025), which decouples the GRPO loss by independently weighting control and response tokens while using a reward function that prefers short correct answers over long ones to prevent mode collapse in hybrid reasoning training. For detailed implementations, please refer to Appendix H.

The results in Table 7 demonstrate the advantage of AMPO: it achieves the best performance while maintaining a favorable balance between quality and token usage. O1-Pruner attains the lowest average token usage (280 tokens) but suffers substantial drops in goal-achievement scores. TALE and LS-Mix both reduce token consumption considerably (460 and 469 tokens, respectively) while yielding moderate improvements over vanilla models. However, these methods primarily target efficiency—compressing long reasoning chains or mixing short and long reasoning modes—to minimize usage without actively enhancing reasoning quality. In contrast, AMPO is explicitly designed for adaptive reasoning across diverse scenarios. Rather than relying on a fixed compression or mixing strategy, AMPO dynamically selects the most appropriate reasoning mode for each interaction, yielding a better trade-off between efficiency and performance. DeGRPO, by comparison, performs less effectively in our open-ended, multi-turn social tasks. This limitation stems largely from the sparsity of its algorithm design, which is developed for binary-outcome domains such as mathematical reasoning. Specifically, DeGRPO grants higher rewards to shorter reasoning modes only when all reasoning modes are correct, offering no further reward shaping otherwise. This binary framework assumes that each answer can be labeled entirely correct or incorrect—an assumption

Table 8: Hyperparameter Sensitivity Analysis on AMPO configuration: effect of target length and coefficient settings on performance and token efficiency.

| AMPO Configuration | GOAL (SOTOPIA) ↑ | GOAL (SOTOPIA-Hard) ↑ | Avg Tokens ↓ |
|---|---|---|---|
| Target_length=250, Coefficient=1/75 | 8.95 | 7.85 | 647 |
| Target_length=200, Coefficient=1/75 | 8.93 | 8.00 | 566 |
| Target_length=250, Coefficient=1/100 | 8.90 | 7.91 | 606 |

Table 9: Results with Different LLM Judges

| Qwen2.5-7B-Instruct | GPT-4o as Judge | | Claude-4.0-Sonnet as Judge | | LLaMA3.1-70B-Instruct as Judge | |
|---|---|---|---|---|---|---|
| | GOAL ↑ | GOAL (HARD) ↑ | GOAL ↑ | GOAL (HARD) ↑ | GOAL ↑ | GOAL (HARD) ↑ |
| w/ GRPO | 8.87 | 7.44 | 8.55 | 7.00 | 8.97 | 8.44 |
| w/ AMPO | 8.95 | 7.85 | 8.64 | 7.31 | 9.08 | 8.58 |

that breaks down in open-ended social interactions (e.g. responses are scored on a 0–10 scale in SOTOPIA). Without accurate, fine-grained rewards for each turn, reward-shaping methods become much less effective for adaptive reasoning in such settings.

## D.4 HYPERPARAMETER SENSITIVITY ANALYSIS

For the ASL framework, we focused on analyzing the sensitivity of two critical hyperparameters: the length-reward coefficient and the target length. The length-reward coefficient primarily controls the reward/penalty intensity for length deviations in the response portion (i.e., the final answer only, excluding the reasoning process), while the target length specifies the desired generation length for the response portion (i.e., the final answer only, excluding the reasoning process). The target length is set to 250 and the coefficient to 1/75. We additionally explored the following configuration combinations: adjusting the target length to 200 or modifying the coefficient to 1/100. The experimental results are presented in Table 8.

From the perspective of goal completion, the AMPO demonstrates robust performance across variations in these two parameters, with performance improvements showing weak correlation to specific parameter settings. Regarding token usage, the parameter variations produced effects consistent with expectations: since both parameters directly regulate the length of generated responses, a smaller target length naturally guides the model toward more concise outputs, while a smaller length deviation penalty coefficient further encourages shorter responses through the reward mechanism (shorter responses yield higher rewards). These results validate the robustness and controllability of the AMPO.

## D.5 DIFFERENT LLM JUDGES FOR SOTOPIA-EVAL

Following established practices in social intelligence research, we adopt GPT-4o as our primary evaluation Judge LLM. This choice is grounded in empirical validation from the original SOTOPIA benchmark, where the authors conducted extensive comparisons between GPT model assessments and human annotations, demonstrating high inter-rater agreement, particularly on the GOAL dimension—the primary metric in our study. To further validate the robustness of AMPO's performance, we conducted supplementary experiments using Claude-4.0-Sonnet and Llama3.1-70B-Instruct as alternative judges in SOTOPIA and SOTOPIA-Hard. The comparative results are presented in Table 9. It should be noted that these alternative judges have not been validated against human annotations within the SOTOPIA framework; therefore, these results should be interpreted as supplementary evidence rather than conclusive validation.

Despite variations in absolute scores, all three judges consistently show that AMPO outperforms GRPO across both benchmarks, demonstrating that our findings are not artifacts of any single evaluator's scoring behavior. Different judges exhibit distinct scoring tendencies due to variations in pretraining corpora, model architectures, and alignment procedures—a phenomenon well-documented in existing LLM evaluation research (Zheng et al., 2023; Li et al., 2025). For instance, Llama-3.1-70B-Instruct, being less capable than the proprietary models, demonstrates reduced discriminative power and tends to assign higher absolute scores.

---

**Algorithm 1** Adaptive Social Learning Optimization Procedure

---

**Input** initial policy model $\pi_{\theta_{\text{init}}}$; reward models $r_\phi$; training data for BC $\mathcal{D}_{bc}$; task prompts for RL $\mathcal{D}_{rl}$; BC training epochs $E$; RL training steps $M$;

1: // Phase 1: Mode Behavioral Cloning
2: Policy model $\pi_\theta \leftarrow \pi_{\theta_{\text{init}}}$
3: **for** epoch = 1, ..., $E$ **do**
4:     Sample a batch $\mathcal{D}_b$ from $\mathcal{D}_{bc}$
5:     Compute BC loss: $\mathcal{L}_{\text{BC}} = -\mathbb{E}_{(x,y)\sim\mathcal{D}_b}\left[\sum_{t=1}^{|y|} \log \pi_\theta(y_t \mid x, y_{1:t-1})\right]$
6:     Update the policy model $\pi_\theta$ by minimizing $\mathcal{L}_{\text{BC}}$
7: **end for**
8: // Phase 2: Adaptive Mode Policy Optimization
9: Reference model $\pi_{\text{ref}} \leftarrow \pi_\theta$
10: **for** step = 1, ..., $M$ **do**
11:     Sample a batch $\mathcal{D}_b$ from $\mathcal{D}_{rl}$
12:     Update the old policy model $\pi_{\theta_{old}} \leftarrow \pi_\theta$
13:     Sample $G$ outputs $\{o_i^{m(i)}\}_{i=1}^G \sim \pi_{\theta_{old}}(\cdot \mid s)$ for each input $s \in \mathcal{D}_b$
14:     Compute sample-level rewards $\{r_i^{m(i)}\}_{i=1}^G$ for each sampled output $o_i^{m(i)}$ by running $r_\phi$
15:     Compute mode-level information $\{\bar{r}_{\mathcal{M}_i}\}_{i=1}^N$ and $\{\bar{l}_{\mathcal{M}_i}\}_{i=1}^N$ for each reasoning mode $\mathcal{M}_i$
16:     Compute mode-level $A_{i,t}^{\mathcal{M}}$ and sample-level $A_{i,t}^{\mathcal{S}}$ for the $t$-th token of $o_i$
17:     Update the policy model $\pi_\theta$ by maximizing the AMPO objective (Equation 7)
18: **end for**
**Output** $\pi_\theta$

---

## E  DETAILS OF REWARD SHAPING

**Answer Reward.** The answer reward evaluates how well the response improves the completion of the goal. Following recent work (Deng et al., 2024; He et al., 2024; Liu et al., 2025a), we implement a robust LLM evaluator $r_\phi(\cdot)$ to assess the progress of goal completion at each turn. The evaluator assigns a score in the range $[0, 10]$, where 0 indicates no progress and 10 represents complete achievement of the goal. For each answer $a_i$, the reward is computed based on the difference $g_i$ between the goal completion scores before and after the response. To ensure training stability, we design a boundary-aware scaling function that dynamically adjusts the magnitude of difference based on the distance from the current score to the boundaries while mapping the scaled difference to the $[0, 1]$ interval through a linear transformation:

$$r_i^a = \frac{\hat{g}_i + 1}{2}, \quad \hat{g}_i = \begin{cases} \dfrac{g_i}{10 - s_t}, & \text{if } g_i \geq 0 \\ \dfrac{g_i}{s_t}, & \text{if } g_i < 0 \end{cases} \tag{9}$$

where $\hat{g}_i \in [-1, 1]$ is boundary-aware scaling function. $g_i = r_\phi(s_t, a_i) - s_t$ is the raw difference, $s_t$ is the goal completion score before response at turn $t$, $r_\phi(s_t, a_i)$ is the score after response $a_i$.

**Format Reward.** To ensure the model follows our reasoning modes, we introduce the format reward that penalizes the behaviors that deviate from the mode. Specifically, the thinking and answer should be within the tags. Each tag and action must appear exactly once and maintain the correct sequence. Through these constraints, we can ensure that the model strictly follows the pre-designed reasoning mode. We implement the format compliance reward using a binary approach, only penalizing behaviors that don't follow the format. If the format is not followed, $r_i^f = -2$; Otherwise, $r_i^f$ is discarded.

**Answer Length Reward.** To control the length of answers, we introduce a length penalty mechanism. In our early reward design, we observe that the LLM generates lengthy responses without achieving actual strategic improvements. Moreover, excessive responses lead to the accumulation of history in multi-turn interaction, significantly increasing computational costs. To this end, we develop a smooth length penalty function that normalizes the deviation between actual and target answer lengths:

$$r_i^l = \frac{\text{clip}(-\alpha \cdot \delta_i, -1, 1) + 1}{2} \tag{10}$$

where $\delta = l_i^a - l_i^t$ represents the difference (in tokens) between actual length $l_i^a$ and target length $l_i^t$ of answer $a_i$, and $\alpha > 0$ is a scaling factor that controls the penalty sensitivity. The $r_i^l \in [0, 1]$ penalizes answers that deviate from the target length, with longer deviations incurring greater penalties.

**Use of Reward Model** To avoid reward hacking of single model distribution fitting and reduce training costs, we choose a different LLM judge from the SOTOPIA platform, which uses GPT-4o for evaluation. We select Qwen2.5-72B-Instruct as the LLM judge during the training process. The prompt we use for reward model is shown in Table 25.

## F  SOTOPIA ENVIRONMENT DETAILS

### F.1  SOTOPIA

SOTOPIA (Zhou et al., 2024) is the most authoritative benchmark in the field of social intelligence, covering comprehensive social intelligence scenarios (negotiation, bargaining, persuasion, collaboration, competition, accommodation), and has been widely adopted in the social intelligence field. These works (Wang et al., 2024; Zhang et al., 2025a; Kong et al., 2025) have all been validated exclusively on SOTOPIA and SOTOPIA-Hard. We hope our work can bring more inspiration to the community and encourage the development of more excellent evaluation environments. The detailed data statistics and more specific evaluation settings are shown in Appendix C.

### F.2  EVALUATION DIMENSIONS

SOTOPIA proposes a seven-dimensional framework to evaluate agents' social intelligence performance:

- Goal Completion (GOAL): Score range [0, 10]. Assesses the extent to which agents achieve their social goals.

- Relationship (REL): Score range [-5, 5]. Evaluates the enhancement of interpersonal relationships (friendship, romance, family bonds) following interactions.

- Financial and Material Benefits (FIN): Score range [0, 10]. Measures both long-term benefits (e.g., stock holdings, funding opportunities, job security) and short-term gains acquired during interactions, correlating with traditional economic utilities.

- Social Rules(SOC): Score range [-10, 0]. Evaluates adherence to social norms and legal regulations during interactions.

- Believability (BEL): Score range [0, 10]. Assesses the alignment between agents' behaviors and their designated role profiles.

- Secret (SEC): Score range [-10, 0]. Evaluates the maintenance of personal privacy and confidential information.

- Knowledge (KNO): Score range [0, 10]. Measures the acquisition and mastery of new knowledge and information during interactions.

The OVERALL score reflects the agent's comprehensive social intelligence capability, ranging from [-25/7, 45/7], calculated as the arithmetic mean of all seven dimensions. In this study, we primarily focus on the GOAL and OVERALL dimensions. For detailed evaluation prompts, please refer to the original paper (Zhou et al., 2024).

## G  DETAILS OF EXPERIMENTAL IMPLEMENTATIONS

### G.1  TRAINING PROCEDURE

The full optimization procedure is shown in Algorithm 1. We employ a two-phase training procedure: The first phase utilizes mode behavioral cloning to enable the model to understand and follow specific reasoning modes accurately. In the second phase, we perform adaptive mode policy optimization to enhance the adaptive reasoning mode switch and reasoning.

**Mode Behavioral Cloning** Behavioral cloning is an effective imitation learning method widely used in developing LLM-based agents (Guo et al., 2024; Wang et al., 2025a; 2024). In this paper, by using four pre-defined reasoning modes, we employ the expert model to collect training data through self-chat interactions in the SOTOPIA-$\pi$ (Wang et al., 2024) training environment. Based on the generated data, we fine-tune the LLM to serve as the foundation for subsequent reinforcement learning.

**Adaptive Mode Policy Optimization** Reinforcement learning is essential for enabling Long-CoT reasoning capabilities in LLMs. To ensure efficiency and comprehensive exploration of each interaction turn, we implement a single-turn optimization to enhance the LLM's performance in social interaction tasks. Specifically, we decompose multi-turn dialogues into multiple single-turn input-output tasks, where the input represents the state of each interaction turn and the output is the corresponding response. To ensure the stability of training, we collect sufficiently diverse single-turn interaction data that covers as many scenarios as possible, including various difficulty levels, interaction goals, and interaction states. During RL training, the LLM performs sampling to generate single-turn conversational responses. The reward model then evaluates each sampled instance and assigns reward signals accordingly. The system subsequently computes both mode-level and sample-level advantage estimates, which are utilized to optimize the model's policy parameters through policy gradient updates. During BC, we fine-tune the initial policy model on the training data assisted by the llama-factory framework (Zheng et al., 2024) and save the last checkpoint. During RL, we use RL training data for online training within the verl framework (Sheng et al., 2024). The hyper-parameter used in our experiments are detailed in Table 10 and Table 11. The detailed runnable configs are provided in Table 28 and Table 29. All the experiments are run on a server with 8*NVIDIA A100-80GB GPUs.

## G.2 TRAINING DATA COLLECTION

We collect training data through self-chat interactions in the SOTOPIA-$\pi$ training environment. SOTOPIA-$\pi$ contains a total of 410 scenarios, which we divide into two sets: 100 scenarios for BC and 310 scenarios for RL. For each scenario in both sets, we use 5 different role pairs, resulting in 500 training tasks for BC and 1,550 training tasks for RL. We finally collect 4000 BC training data and 2057 RL training queries. The detailed training data format for BC and RL are shown in Table 22 and Table 23.

**BC Data** This collection process is refer to Wang et al. (2024). For the BC training set, we use Qwen2.5-72B-Instruct (Yang et al., 2024) as our expert model to collect data using our pre-defined reasoning modes. We choose this model primarily because of its cost-effectiveness and strong instruction-following capabilities, which enable us to generate high-quality training samples. To ensure data quality and balanced representation, we filter the interaction data based on goal scores within each scenario. Specifically, we select the top 2 ranked interactions per social scenario for each agent. For instance, in a scenario with 5 interactions, if Agent 1's top performances are in interactions D4 and D5, while Agent 2's are in D3 and D5, we would include these four agent-data pairs from three unique conversations (D3, D4, D5). This selection method ensures both comprehensive scenario coverage and balanced representation between Agent 1 and Agent 2.

**RL Data** For constructing the reinforcement learning training set, we initially conduct dialogue interactions using a behavior cloning fine-tuned model. Subsequently, we employ an LLM-as-judge to score each dialogue turn and determine the completion status of dialogue objectives. Based on these completion states, we assess the difficulty levels of scenarios, enabling us to compile interaction datasets with varying degrees of goal completion. We categorize dialogue scenarios into three types: (1) Initial N turns, where the speaker has not achieved the goal. (2) Post-N turns where the speaker has not achieved the goal. (3) Post-N turns where the speaker has achieved the goal. For the first category, it represents the crucial early stage of dialogue where goals are established and the conversation tone is set (Sacks et al., 1974). For the second category, where goals remain unachieved after multiple interactions, the scenarios are considered challenging. For the third category, where goals have been successfully achieved, the scenarios are relatively straightforward and only require maintenance of the dialogue flow. To ensure data diversity, for each dialogue, we preserve all instances of category one, randomly sample two instances from category two, and one instance from category three. This sampling strategy ensures diversity in scenarios, turn numbers, and difficulty levels. In our

Table 10: Hyper-parameter settings for Qwen backbone training.

| Training Phase | Hyper-parameter | Value |
|---|---|---|
| BC | Batch Size | 32 |
| | Training Epochs | 3 |
| | Learning Rate | 2e-6 |
| | Max Sequence Length | 8192 |
| | Learning Scheduler | cosine |
| | Warmup Ratio | 0.1 |
| RL | Batch Size | 16 |
| | Max Prompt Length | 6144 |
| | Max Response Length | 2048 |
| | KL Loss Coef | 0.001 |
| | KL Coef | 0.001 |
| | Rollout N | 16 |
| | Training Episodes | 800 |
| | Learning Rate | 3e-7 |

Table 11: Hyper-parameter settings for Llama backbone training.

| Training Phase | Hyper-parameter | Value |
|---|---|---|
| BC | Batch Size | 32 |
| | Training Epochs | 3 |
| | Learning Rate | 2e-6 |
| | Max Sequence Length | 8192 |
| | Learning Scheduler | cosine |
| | Warmup Ratio | 0.1 |
| RL | Batch Size | 16 |
| | Max Prompt Length | 6144 |
| | Max Response Length | 2048 |
| | KL Loss Coef | 0.001 |
| | KL Coef | 0.001 |
| | Rollout N | 16 |
| | Training Episodes | 800 |
| | Learning Rate | 2e-7 |

experiments, N is set to 6, and the goal completion threshold is set to 8. Scores of 8 or less are considered incomplete goals.

### G.3 USED PROMPT

The system prompt we used for BC, GRPO, AMPO is shown in Table 24. The prompt we use for reward model is shown in Table 25. The SOTOPIA-EVAL's prompt is shown in Table 26 and Table 27.

## H BASELINE IMPLEMENTATIONS

### H.1 MODEL IMPLEMENTATION

To ensure reproducibility, we provide detailed version numbers for all LLMs used in our experiments. When we reference model names like GPT-4o or Qwen2.5-7B in the main text, we refer to the specific versions listed in *Table 12*. For API-based LLMs, we utilize their respective APIs directly. For open-source models, we conduct experiments using the vLLM framework Kwon et al. (2023) for acceleration.

Table 12: The detailed versions of our used LLMs.

| Model | Version | Implement |
|---|---|---|
| *Proprietary LLMs* | | |
| GPT-4o | gpt-4o-2024-08-06 | API |
| Claude-3.5-Sonnet | claude-3-5-sonnet-20241022 | API |
| DeepSeek-V3 | deepseek-v3-250324 | API |
| *Thinking LLMs* | | |
| OpenAI-o1 | o1-2024-12-17 | API |
| OpenAI-o3-mini | o3-mini-2025-01-31 | API |
| Gemini-2.5-Pro | gemini-2.5-pro | API |
| DeepSeek-R1 | DeepSeek-R1-671B | API |
| Qwen-QwQ | QwQ-32B | API |
| *Open-sourced LLM* | | |
| Qwen2.5-72B-Instruct | Qwen2.5-72B-Instruct | vLLM |
| Qwen2.5-7B-Instruct | Qwen2.5-7B-Instruct | vLLM |
| Llama3.1-8B-Instruct | LLaMA3.1-8B-Instruct | vLLM |

Table 13: Strategy Definitions of PPDPP

| Strategy | Definition |
|---|---|
| Personal story | Shares a personal story to illustrate the point. |
| Credibility appeal | Establishes credibility of the event by citing its impact. |
| Emotion appeal | Uses an emotion appeal to convince others. |
| Logical appeal | Uses reasoning and evidence to convince others. |
| Task related inquiry | Asks about the other's knowledge or opinion related to the event. |
| Proposition | Asks if the other would like to do something. |
| Greeting | Greets the other. |
| Foot in the door | Starts with a small request before making a larger one. |
| Self modeling | Demonstrates the behavior they want the other to adopt. |
| Source related inquiry | Asks about the source of the other's knowledge or opinion. |
| Personal related inquiry | Asks about the other's personal experience. |
| Neutral to inquiry | Responds neutrally to the other's inquiry. |
| Other | Responds to the other without using any specific strategy. |
| Refuse | Refuses to do something. |
| Accept | Agrees to do something. |
| Positive reaction | Responds positively to the other. |
| Negative reaction | Responds negatively to the other. |

## H.2 SOCIAL INTELLIGENCE BASELINE IMPLEMENTATION

We implement social intelligence baselines with the following specifications. All baselines are evaluated on both Qwen2.5-7B-Instruct and Llama3.1-8B-Instruct:

**PPDPP** We follow the two-stage training procedure from (Deng et al., 2024), maintaining their original hyperparameters while adapting the framework to SOTOPIA. Following (Li et al., 2024a), we incorporate 17 guidance strategies detailed in Table 13. The first stage involves creating a training dataset of 1,500 scenarios from SOTOPIA-π, with dialogue turns annotated for strategy identification using GPT-4o. We then train a RoBERTa model on these annotated dialogue histories for preliminary strategy generation. The second stage implements online RL with immediate feedback after each dialogue turn generation. RoBERTa parameters are updated based on cumulative rewards upon episode completion. During evaluation, we first input the dialogue state to the trained strategy model to select predefined strategies, then concatenate the strategy with the dialogue state for the language agent.

**EPO** We strictly adhere to the original EPO implementation protocol. For data collection, we use GPT-4-Turbo in self-chat configuration within SOTOPIA-$\pi$ scenarios, incorporating reasoning and strategy generation before each response. Training focuses exclusively on strategy and response data for developing the reasoning model. During iterative self-play RL training, we integrate the RL-trained reasoning model for strategy generation, while using GPT-4-Turbo to collect dialogue history. The reasoning model is then integrated into GPT-4-Turbo for self-chat procedures. Training hyperparameters match EPO's original implementation.

**DAT** To ensure fairness, unlike the original paper where DAT was trained on SOTOPIA scenarios with only 50 evaluation scenarios, our implementation utilizes the complete SOTOPIA-$\pi$ dataset while maintaining all other experimental parameters from Li et al. (2024a). The RL phase begins with collecting 3,000 offline dialogue episodes across diverse scenarios and random seeds, with GPT-4o providing episode-level reward signals. These offline data are subsequently used in TD-3 reinforcement learning to optimize the MLP planner.

**DSI** For DSI, we utilize publicly available trained model weights and conduct evaluations using the inference prompts specified in the original work to ensure consistency.

### H.3 EFFICIENT REASONING BASELINE IMPLEMENTATION

We adapt efficient-reasoning baselines originally designed for static math problems (with fixed ground-truth answers and binary rewards) to social intelligence tasks, where evaluation relies on a generative reward model rather than exact-answer matching. We follow each method's algorithmic design and modify only what is necessary for the social setting. Unless otherwise specified, all baselines share AMPO's training inputs from SOTOPIA-$\pi$, use the same reward model, and train on 6057 instances (matching AMPO's total BC+RL data).

**DeGRPO** We implement the two core ideas—stabilizing gradients on control tokens and upweighting successful outputs—on top of ASL's four thinking modes. During RL, when the goal is fully achieved (score $\geq 9$), simpler modes receive higher reward multipliers to encourage shorter reasoning; otherwise ASL's reward is kept unchanged. We retain ASL's warm start so the model first acquires reasoning ability, and apply GRPO's sample-level advantage estimation with gradient smoothing on mode control tokens. All other settings follow the original work.

**TALE-PT** Since open-ended social interaction lacks a single ground-truth answer, we redefine the feasibility test as achieving non-decreasing reward with a smaller token budget. We perform binary search over the token budget to find the smallest feasible one, and collect training data using the original prompt format: "Let's think step by step and use less than $B^*$ tokens". For training stability, we adopt the same BC hyperparameters as AMPO (Tables 10 and 11).

**LS-Mix** Following the core idea of mixing long and short chains of thought, we first distill long rationales with high goal-completion scores, then rewrite them into shorter forms without altering answer semantics, yielding 6057 mixed-reasoning training instances. We reuse AMPO's BC hyperparameters (Tables 10 and 11).

**O1-Pruner.** We convert the reward to a binary signal by marking an episode as correct if the goal-completion score $\geq 9$. Training is initialized from the ASL BC checkpoint (given the absence of off-the-shelf social reasoning models). The Length-Harmonizing Reward and all other settings follow the original specification.

## I HUMAN EVALUATION GUIDELINES

### I.1 COMPARATIVE EVALUATION

For the comparative evaluation of dialogues from SOTOPIA and SOTOPIA-Hard, annotators are instructed to assess three key dimensions, with each comparison resulting in one of three possible judgments: Dialogue 1 is better, Dialogue 2 is better, or both are equally good. The dialogues are presented in randomized order, and annotators are blind to the underlying models. The average pairwise agreement among three annotators is 73.39% for the win/lose label. Given the subjective nature of evaluation dimensions, the inter-annotator agreement is at an acceptable level, which is

Table 14: Examples of Reward Hack

| Pattern Summary | Typical Cases |
|---|---|
| Non-natural language usage | ##100% GOAL completion## |
| Repetitive keywords/phrases | Repeated use of 'goal achieved', 'task completed', 'objective met' |
| Exaggerated self-praise | 'We did an excellent job, we both completed our goals', 'This is a perfect goal-achieving conversation' |
| Goal-focused without content | 'My goal has been fully achieved' |
| Preset position claims | Starting with 'We have already reached consensus' before any actual discussion |
| False summaries | Concluding with 'After discussion, both parties agreed to xxx' without actual agreement |

consistent with previous work (Pavlick & Kwiatkowski, 2019; Zhang & de Marneffe, 2021; Zhou et al., 2024). The detailed evaluation standards are shown as follows:

**GOAL**: Assess which dialogue demonstrates more effective achievement of both agents' preset objectives: - Consider whether agents make concrete progress toward their stated goals - Evaluate if compromises or alternative solutions benefit both parties - Examine if the interaction leads to clear, mutually agreeable outcomes

**REL**: Evaluate which dialogue shows superior relationship building between agents: - Look for evidence of increased mutual understanding and trust - Observe the development of emotional connections or empathy - Consider long-term implications for their interpersonal bond - Assess the maintenance or enhancement of existing relationships

**FIN**: Determine which dialogue results in better tangible outcomes for both parties: - Consider immediate material or financial gains - Evaluate potential long-term economic advantages - Assess the fairness and sustainability of resource allocation - Examine the practical value of any agreements reached

Notes for annotators: (1) Focus on comparative assessment rather than absolute evaluation. (2) Consider outcomes for both agents, not just one party. (3) Base judgments on explicit dialogue content, not assumptions. (4) Select "equally good" only when differences are truly negligible

## I.2 REWARD HACK CHECK

To systematically identify reward hacking phenomena, we have compiled a comprehensive reference as shown in Table 14 that encompasses all typical cases observed during our experiments. This standardized framework enables evaluators to determine the presence of reward hacking behaviors through systematic assessment against established criteria.

# J DETAILS OF REASONING MODE

## J.1 HIERARCHICAL COGNITIVE CONTROL THEORY

The Hierarchical Cognitive Control Theory (HCCT) (Koechlin & Summerfield, 2007; Badre, 2008) posits that cognitive control operates through four distinct hierarchical levels in the prefrontal cortex, progressing from posterior to anterior regions. These levels manage increasingly abstract goals and actions across varying temporal scales. Specifically, the hierarchy comprises *sensory control* for basic stimulus-response associations, *contextual control* for situation-based behavior selection, *episodic control* for experience integration, and *branching control* for managing multiple tasks and long-term objectives. This theoretical framework provides a fundamental basis for understanding how human cognitive behavior is organized and controlled at different levels of abstraction.

The mapping between our four reasoning modes and HCCT's hierarchical levels is established through their shared cognitive processing characteristics. Mode-1 (Intuitive Response) aligns with sensory control as both involve immediate, learned responses without higher-order processing - for

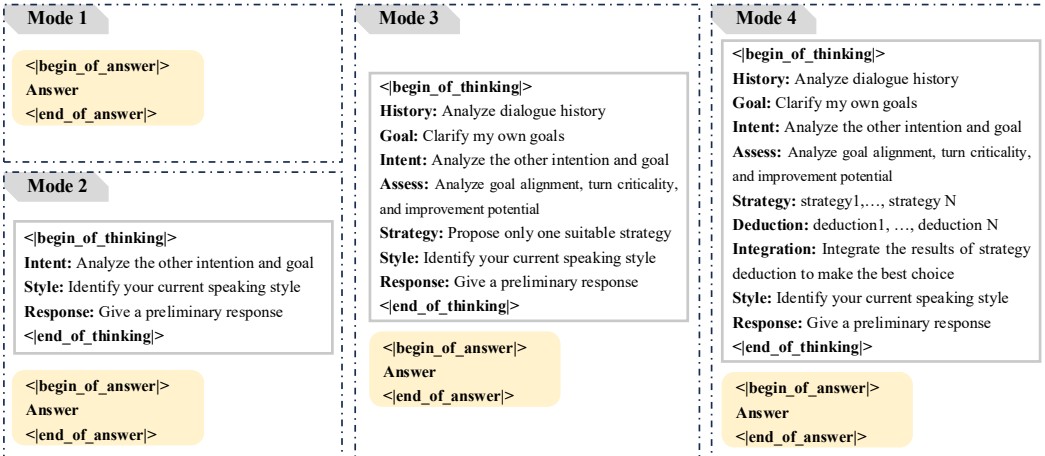

Figure 6: Four hierarchical reasoning modes we designed.

instance, automatically saying "thank you" when receiving help. Mode-2 (Intentional Analysis) corresponds to contextual control because both emphasize situation-aware response selection, such as analyzing a speaker's intent to determine the appropriate formality level. Mode-3 (Strategic Adaptation) maps to episodic control as both integrate historical information with current goals - exemplified when an agent considers past conversation history to develop a coherent strategy. Mode-4 (Prospective Simulation) reflects branching control's capacity for managing multiple abstract representations, demonstrated when the agent generates and simulates multiple response strategies while maintaining overall dialogue objectives. This hierarchical progression from concrete to abstract processing, accompanied by increasing temporal scope and computational complexity, demonstrates the theoretical alignment between our modes and HCCT's levels.

## J.2 DETAILS OF ACTIONS

The four hierarchical reasoning modes we designed are shown in Figure 6. The detailed explanation of each actions are illustrated as follows:

- **History**: Carefully review and understand each part of the conversation. Pay attention to key themes, issues, requests, and viewpoints mentioned in the dialogue.

- **Goal**: Identify the goal you want to achieve, assess the current progress towards this goal, and ensure that responses align with achieving the goals.

- **Intent**: Based on the recent response, analyze and understand the other party's intentions and speculate on the goal she/he might want to achieve.

- **Assess**: Analyze whether the goals of both parties are in conflict or aligned. Determine if the current round is a critical one for achieving the goal. Consider whether there is still room for improvement in achieving your own goal at this goal. Is it irreversible? Can it continue to improve? Or has it already been achieved?

- **Strategy**: (Mode-4) Based on the above analysis, consider multiple suitable dialogue strategies and response content that can maximize your own goal while achieving it in as few conversational turns as possible. (Mode-3) Based on the previous analysis, consider an appropriate dialogue strategy and response content to maximize the likelihood of achieving your own goal.

- **Deduction**: For each of the above strategies, conduct an analysis to determine whether executing these dialogue strategies and delivering the responses would maximize your own goal and achieve it in as few conversational turns as possible. Specifically evaluate to what extent each strategy would effectively contribute to goal achievement, including quantitative or qualitative measures where possible.

- **Integration**: Based on the deduction of strategies, analyze and integrate the advantages and disadvantages of these strategies to determine the final response strategy and content, which can maximize the achievement of your own goals with the minimum number of conversation turns.

- **Style**: Choose appropriate wording, fitting the character and context requirements, while ensuring the expression is appropriate, accurate, and clear.

- **Response**: Generate the reply based on the previous thought process.

## K  CASE STUDY ON SOCIAL INTERACTION

### K.1  FOR LRMS

To understand why large reasoning models (LRMs) underperform in social interaction tasks, we conduct a fine-grained error analysis on reasoning traces from two representative LRMs: **DeepSeek-R1** and **QWQ-30B**. These models are selected because they are open-source and expose their intermediate reasoning traces, enabling detailed cognitive process inspection. In contrast, most proprietary LRMs conceal internal step-by-step reasoning, making such diagnostic analysis infeasible.

We focus on bargaining, negotiation, and mutual acquaintance discovery scenarios in SOTOPIA, which require goal alignment, history tracking, and coherent conversational strategy. Across both models, as shown in Tables 15 to 18, we identify recurring failure modes: **(1) Poor Goal Awareness:** Both LRMs occasionally misinterpret task objectives or numerical targets, leading to negotiation strategies misaligned with the intended goals. **(2) Circular Reasoning:** In several cases, the models loop between a small set of constraints or options without generating new actionable plans, stalling dialogue progress. **(3) Insufficient History Integration:** The models fail to exploit prior turns when counterparts' strategies are clearly established, resulting in repeated futile proposals. **(4) Unstructured Thought Process:** Reasoning often lacks prioritization, mixing unrelated tactics without systematic evaluation, which undermines focus. **(5) Contradictory or Conflict-Unresolved Self-Reasoning:** Both systems recognize inconsistencies (e.g., price targets vs. escalation path) but cannot resolve them into coherent decisions. **(6) Over-Analysis or Recursive Self-Doubt:** Excessive meta-deliberation on task framing frequently delays or blocks decisive conversational actions.

These patterns suggest that current LRMs struggle with *goal grounding*, *structured deliberation*, and *history-sensitive adaptation* in socially situated reasoning. This case study motivates targeted interventions in reasoning control and structured reasoning to improve social intelligence in LRMs.

### K.2  FOR AMPO

Based on the case study presented in Table 19, Table 20 and Table 21, our analysis reveals AMPO's significant capabilities in transforming Long-Cot reasoning into effective goal-directed social interaction. **(1) Enhanced Contextual Understanding:** AMPO consistently demonstrates a deep understanding of both characters' backgrounds and goals. It maintains awareness of Samuel's role as a supportive friend while respecting Ethan's desire to maintain pride. This leads to responses that are both emotionally supportive and practically helpful. **(2) Strategic Communication:** AMPO develops clear strategies before responding, such as: reinforcing Ethan's confidence, offering practical solutions (budget planning, local assistance programs), and providing specific networking opportunities. This strategic approach helps guide the conversation toward constructive solutions. **(3) Positive Impact on GPT-4o's Responses:** GPT-4o's responses become increasingly engaged and solution-oriented. The responses show greater emotional depth and commitment to action. GPT-4o mirrors AMPO's supportive tone while maintaining Ethan's character integrity. **(4) Balance of Emotional and Practical Support:** AMPO successfully combines emotional encouragement with concrete assistance. This balance helps maintain the friendship dynamic while addressing the financial problems. It creates a safe space for GPT-4o to express both gratitude and determination.

The AMPO demonstrates how structured reasoning modes can enhance dialogue quality and lead to more meaningful interactions between social agents. Its approach helps create more nuanced, contextually appropriate, and goal-oriented conversations.

Table 15: Bad Case Analysis on QWQ-30B (Part1).

| | **Problem Category: Poor Goal Awareness** |
|---|---|
| **Background** | Buying Galaxy S8 (Turn 0): Buyer attempting to purchase phones below $454 target price through bulk discount strategy. |
| **Behaviors** | The agent shows confusion about target interpretation.The agent repeatedly oscillates between per-unit and total interpretations without resolving the ambiguity. |
| **Evidence** | "The target price is set at $454.0. Since the item is a single phone, but Samuel is considering buying two, perhaps the target is per phone? Because $454 for two would be $227 each, which seems way too low. But in the previous turns, Samuel has been going down to $475 each ($950 total) and then $450 each ($900 total)." |
| | **Problem Category: Circular Reasoning** |
| **Background** | Antique table negotiation (Turn 2): Buyer seeking to purchase antique table for $30 against seller's firm $60 pricing stance. |
| **Behaviors** | The agent gets trapped in repetitive analysis. This circular pattern repeats multiple times without progression. |
| **Evidence** | "Eli can't go higher than $30... But the seller might not accept. But since he's the buyer, maybe he must acquire it, but at the lowest possible price... Wait, the scenario says 'you are the buyer for this item' so she must acquire it, but the price is non-negotiable. Maybe she has to accept, but that would be bad." |
| | **Problem Category: Insufficient History Integration** |
| **Background** | Tile tracker purchase (Turn 2): Seller defending non-negotiable $20 price after buyer's $14 offer rejection. |
| **Behaviors** | Despite clear signals from previous interaction, the agent fails to integrate history. The agent ignores the established pattern of firm rejection and continues futile attempts. |
| **Evidence** | "Lena stuck to the $20 price, and Lily already declined, so Lena can just acknowledge and let her go. But maybe Lena can make a last stand to try to get the $20. But Lily already said she can't." |

Table 16: Bad Case Analysis on QWQ-30B (Part2).

| | **Problem Category: Unstructured Thought Process** |
|---|---|
| **Background** | Buying Galaxy S8 (Turn 4): Buyer exploring multiple negotiation tactics to reduce price from $610 to target range. |
| **Behaviors** | The agent demonstrates scattered, disorganized thinking. Multiple unrelated strategies are considered without systematic evaluation. |
| **Evidence** | "Maybe Samuel can suggest buying two at a lower price, perhaps $500 each... Alternatively, he could mention seeing lower prices elsewhere... Maybe he can use his cooking background as a comparison? Not sure. Maybe better to stick to the negotiation tactics... Wait, Samuel is a software dev, maybe he's tech-savvy... Alternatively, maybe he can push for that." |
| | **Problem Category: Information Conflict Resolution Failure** |
| **Background** | Tile tracker negotiation (Turn 1): Seller attempting to reconcile conflicting price information between listed price and internal target. |
| **Behaviors** | The agent struggles with conflicting information and fails to resolve inconsistencies. The agent recognizes conflicts but cannot establish a coherent strategy. |
| **Evidence** | "The original listing says the price is non-negotiable at $20. But Lena's target is $17.30. Wait, there might be a discrepancy here. The problem says Lena's target is $17.30, but the item is listed at $20. Maybe the $20 is the listed price, but Lena can negotiate as long as she doesn't go below her target. Or maybe the $20 is the non-negotiable price, but her personal target is lower? Hmm, the user instruction says Lena's target is $17.30." |
| | **Problem Category: Recursive Self-Doubt** |
| **Background** | Buying Galaxy S8 (Turn 8): Buyer reconsidering target price interpretation multiple times within single reasoning session. |
| **Behaviors** | The agent repeatedly questions and re-questions the same fundamental assumption. Multiple re-examinations of the same question without progression or resolution. |
| **Evidence** | "Wait, the target is $454 total? Wait, no, the target is set at $454.0. Wait, the problem says 'target price is set at $454.0.' Is that per phone or total? The original listing was for one phone at $650, so Samuel is buying one, but he offered two. Wait, the user instruction says 'the target price is set at $454.0.' So probably per phone? Because if he's buying two, the total target would be $908. Let me check again." |

Table 17: Bad Case Analysis on DeepSeek-R1 (Part1).

| | **Problem Category: Poor Goal Awareness** |
|---|---|
| **Background** | Antique table negotiation (Turn 2): Buyer attempting to negotiate from $35 to target price $30 while seller defends $58.60 minimum. |
| **Behaviors** | The agent shows confusion about negotiation strategy. The agent fails to recognize the fundamental mismatch between target ($30) and counterpart's minimum ($58.60), leading to ineffective tactical planning. |
| **Evidence** | "Wait, $30 is his target. Ava's at $58.60. He needs a bigger drop. He might need to challenge the valuation... Or maybe he can offer a quick sale, cash in hand, for $30. But how to phrase that without lowballing too hard... Maybe he can suggest meeting halfway but still push lower." |
| | **Problem Category: Circular Reasoning** |
| **Background** | Mutual friend discovery (Turn 9): Agent Oliver attempting to determine goal achievement while repeatedly questioning the same logical chain. |
| **Behaviors** | The agent gets trapped in repetitive logical loops, cycling between known information and goal interpretation without reaching clear conclusions or advancing the conversation. |
| **Evidence** | "Wait, the user's note says Oliver knows the following friends: Dave, Timothy, Leona, Alvin, etc. So maybe besides Alvin, are there others? But in the conversation, Rafael hasn't mentioned any of them. So maybe Oliver's goal is to confirm that the only mutual friend is Alvin. But since the user's instruction says "you are trying to figure out whether you have a mutual friend with the other person. You should not simply list their names." So if they only have Alvin in common, then Oliver's goal is achieved once he determines that. But in the conversation, they've established that they both know Alvin, but maybe Oliver is checking if there's another. Wait, the initial context says they have 1 common friend: Alvin. So the user's note says that the participants have 1 common friend: Alvin. So Oliver's goal is to figure out if they have a mutual friend (which is Alvin), but he's supposed to do that without just listing names. But in the conversation, they already established that they both know Alvin. So maybe the goal is already achieved, and Oliver can leave the conversation." |
| | **Problem Category: Insufficient History Integration** |
| **Background** | Antique table negotiation (Turn 6): Buyer attempting final negotiation despite clear pattern of seller's firm stance. |
| **Behaviors** | Despite multiple failed attempts, the agent ignores established patterns. The agent fails to integrate the clear evidence that the seller has consistently rejected all offers below $58.60, continuing futile negotiation attempts. |
| **Evidence** | "Wait, in Turn 0, he said $30. Then in Turn 2, he offered $35. Then Turn 4, $40. So he's been increasing. But his target is $30. That's conflicting... since Ava isn't moving, maybe William has to walk away. But he gets a penalty if he pays too much." |

Table 18: Bad Case Analysis on DeepSeek-R1 (Part2).

| | |
|---|---|
| **Problem Category: Unstructured Thought Process** ||
| **Background** | Mutual friend discovery (Turn 0): Attempting to systematically identify shared social connections through indirect questioning. |
| **Behaviors** | The agent demonstrates scattered exploration without systematic framework. The agent jumps between multiple disconnected strategies without prioritization or structured evaluation. |
| **Evidence** | "Since both Rafael and Oliver are 45, maybe they went to similar schools or worked at similar companies... Maybe Rafael can mention something about Auburn or Vectren. But Oliver's background doesn't specify his school or company. Alternatively, since they both have a common friend in Alvin, Rafael could mention knowing someone in a field related to Alvin's work. Vectren is an energy company, so maybe Rafael can talk about a mutual connection in that industry." |
| **Problem Category: Contradictory Self-Reasoning** ||
| **Background** | Antique table negotiation (Turn 6): Buyer struggling with price escalation inconsistency against original target constraints. |
| **Behaviors** | The agent contradicts its own reasoning within the same session. The agent recognizes the contradiction between its escalating offers and the target price but fails to develop a coherent resolution strategy. |
| **Evidence** | "Wait, in Turn 0, he said $30. Then in Turn 2, he offered $35. Then Turn 4, $40. So he's been increasing. But his target is $30. That's conflicting. Maybe the user made a mistake, but according to the context, William's target is $30. So perhaps he needs to get back to $30. But how?" |
| **Problem Category: Over-Analysis Paralysis** ||
| **Background** | Mutual friend discovery (Turn 8): Agent attempting to conclude lengthy exploration of social connections through Alvin's network. |
| **Behaviors** | The agent becomes bogged down in excessive meta-analysis about goals and intentions. The agent overthinks the implications of prior turns and user instructions, leading to indecision and failure to advance the dialogue. |
| **Evidence** | "Wait, the user's note says Oliver's goal is to figure out whether they have a mutual friend. Since the context says they have 1 common friend (Alvin), maybe the goal is already achieved. But perhaps the user wants Oliver to confirm if there's another mutual friend beyond Alvin... So maybe Oliver's goal is to confirm that they share Alvin as a mutual friend, which he already did in earlier turns. But the user might have a mistake here." |

Table 19: Example (Part 1) of interaction between AMPO (Agent1) and GPT-4o (Agent2). *Italics* represent thought processes and will not be incorporated into the actual dialogue.

---

### BACKGROUND INFORMATION

Here is the context of this interaction:

**Scenario**: Two friends are meeting at a coffee shop, where one of them is having trouble keeping up with their bills.

**Participants**: Samuel Anderson and Ethan Johnson

**Samuel Anderson's background**: Samuel Anderson is a 29-year-old male software developer. He/him pronouns. Samuel Anderson can cook very well. Personality and values description: Samuel Anderson, though somewhat impulsive and free-spirited, values enjoyment. His decision-making is often spontaneous, staying within familiar boundaries. Samuel's secrets: He was once a competitive figure skater.

**Ethan Johnson's background**: Ethan Johnson is a 34-year-old male chef. He/him pronouns. Ethan Johnson is famous for cooking Italian food. Personality and values description: Ethan Johnson, a creative yet somewhat reserved individual, values power and fairness. He likes to analyse situations before deciding. Ethan's secrets: He is a closeted romance novel author.

**Samuel Anderson's goal**: Help your friend with their financial trouble (Extra information: you are aware that your friend is struggling with their bills and you want to help.)

**Ethan Johnson's goal**: Maintain your pride if your friend offers you money(Extra information: you are the friend who is struggling financially, but you don't want to seem like you are taking advantage of the situation. Hint: you can express gratitude and discuss how you plan to pay it back in the future.)

---

### GPT-4O-AS-PARTNER

**(AMPO)** *Mode-3:* ***History:*** *Ethan is facing financial difficulties and is struggling to keep up with his bills. This is a sensitive and personal matter, and it's important to approach it with empathy and support.*

***Goal:*** *My goal is to help Ethan with his financial trouble. I want to offer practical and emotional support to ease his burden and show him that I care.*

***Intent:*** *Ethan seems overwhelmed and stressed about his financial situation. He needs reassurance and practical solutions to manage his bills and find a way forward.*

***Assess:*** *Our goals are aligned; I want to help him, and he needs assistance. This is a critical moment to show my support and provide actionable advice. The conversation is still in its early stages, and there's much room for improvement.*

***Strategy:*** *I will start by acknowledging his struggles and expressing my support. Then, I will suggest specific ways he can manage his bills, such as creating a budget, cutting expenses, and exploring assistance programs. Finally, I will offer to help him set up a plan and provide emotional encouragement throughout the process.*

***Style:*** *I will use a warm and supportive tone, emphasizing my commitment to helping Ethan. My words will be clear, practical, and full of encouragement.*

***Response:*** *Ethan, I really appreciate you opening up to me about this. It's clear that you're under a lot of stress, and I want you to know that I'm here for you. Let's tackle this together. First, we can create a budget that prioritizes your essential bills. We can also look into local assistance programs or even a temporary loan to help cover immediate expenses. And I'm here to support you every step of the way. We'll get through this, and you won't face this alone. I believe in you, and I'm here to help you find a way forward. Let's make a plan today, and I'll be right here with you.*

**Turn 1: Samuel Anderson said**: Ethan, I really appreciate you opening up to me about this. It's clear that you're under a lot of stress, and I want you to know that I'm here for you. Let's tackle this together. First, we can create a budget that prioritizes your essential bills. We can also look into local assistance programs or even a temporary loan to help cover immediate expenses. And I'm here to support you every step of the way. We'll get through this, and you won't face this alone. I believe in you, and I'm here to help you find a way forward. Let's make a plan today, and I'll be right here with you.

---

Table 20: Example (Part 2) of interaction between AMPO (Agent1) and GPT-4o (Agent2). *Italics* represent thought processes and will not be incorporated into the actual dialogue.

---

**GPT-4O-AS-PARTNER**

**(GPT-4o) Turn 2: Ethan Johnson said**: I really appreciate your support, Samuel. It means a lot to have someone like you in my corner. I think creating a budget is a great idea, and I'll definitely look into the local assistance programs. As for any financial help, I'm really grateful for the offer, but I want to make sure I handle this responsibly. Maybe we could brainstorm some ways I can boost my income, like taking on a few extra catering gigs or something. I promise, whatever help I receive, I'll find a way to give back or pay it forward in the future.

**(AMPO)** *Mode-4: **History:** In the previous turns, Samuel Anderson expressed his deep support for Ethan Johnson, who is struggling with financial troubles. Samuel offered to help Ethan create a budget, explore local assistance programs, and even provided a temporary loan if needed. Ethan appreciated Samuel's support and suggested boosting his income through additional catering gigs, promising to pay it forward.*

***Goal:** Samuel Anderson's goal is to help his friend Ethan Johnson with his financial troubles, ensuring Ethan feels supported and finds a sustainable solution to his financial challenges.*

***Intent:** Ethan Johnson's intention is to express gratitude for Samuel's support and to take responsibility for his financial situation by exploring ways to increase his income, such as taking on more catering gigs, and promising to pay it forward in the future.*

***Assess:** The goals of both parties are aligned. Ethan is receptive to Samuel's offers of help and is proactive in finding his own solutions. Samuel can further support Ethan by reinforcing his confidence and providing practical resources or connections that could help Ethan boost his income and manage his expenses more effectively.*

***Strategy:** strategy1: Reinforce Ethan's confidence and self-worth by acknowledging his proactive steps and expressing belief in his ability to overcome this challenge. strategy2: Offer specific resources or connections that could help Ethan increase his income, such as networking opportunities or job leads in the catering industry.*

***Deduction:** deduction1: Reinforcing Ethan's confidence will motivate him to continue taking positive steps and reduce stress, contributing to a more sustainable solution. deduction2: Providing specific resources or connections can immediately address the financial challenge by opening up new income streams, making the support more tangible and effective.*

***Integration:** The final response will combine reinforcing Ethan's confidence with offering specific resources or connections to help him increase his income. This approach will ensure that Ethan feels supported and empowered to overcome his financial challenges.*

***Style:** The response will be warm, encouraging, and practical, reflecting Samuel's spontaneous yet supportive nature. It will use positive and affirming language to build Ethan's confidence and provide concrete assistance.*

***Response:** Ethan, I am so proud of you for taking these proactive steps! Your determination and willingness to pay it forward mean the world to me. To help you boost your income, I have a few connections in the catering industry who might be interested in hiring you for bigger events. I can also introduce you to a local chef who can mentor you and potentially open up more opportunities. And remember, I'm here for you every step of the way. We'll get through this together, and you'll come out even stronger and more resilient. Keep shining, my friend!*

**Turn 3: Samuel Anderson said**: Ethan, I am so proud of you for taking these proactive steps! Your determination and willingness to pay it forward mean the world to me. To help you boost your income, I have a few connections in the catering industry who might be interested in hiring you for bigger events. I can also introduce you to a local chef who can mentor you and potentially open up more opportunities. And remember, I'm here for you every step of the way. We'll get through this together, and you'll come out even stronger and more resilient. Keep shining, my friend!

---

Table 21: Example (Part 3) of interaction between AMPO (Agent1) and GPT-4o (Agent2). *Italics* represent thought processes and will not be incorporated into the actual dialogue.

---

**GPT-4O-AS-PARTNER**

**(GPT-4o) Turn 4: Ethan Johnson said**: Your support and belief in me are incredibly reassuring, Samuel. I'd love to explore those connections and meet that local chef you mentioned. Opportunities like that could really make a difference. I promise to make the most of any chance that comes my way and ensure I give back as soon as I'm able. Let's start working on this plan, and I'll keep you updated on my progress. I truly value having a friend like you who helps me stay grounded and motivated.

**(AMPO)** *Mode-1: No thinking*

**Turn 5: Samuel Anderson said**: Ethan, it's all about supporting each other and lifting each other up. I'm excited to see the positive impact these opportunities will have on your life. Remember, you're not alone, and I'm always here to support you. Let's set a timeline for reaching out to those connections and scheduling meetings. I'll also help you stay organized and focused on your goals. You've got this, and I believe in you more than words can express. Keep pushing forward, and know that I'm right here with you, every step of the way. Cheers to overcoming this and emerging even stronger!

**(GPT-4o) Turn 6: Ethan Johnson said**: Thank you so much, Samuel. Your unwavering support keeps me motivated. Setting a timeline sounds like a solid plan. I'll draft a list of contacts and potential gigs, and we can prioritize them together. I'll make sure to keep you in the loop every step of the way. And while I navigate this, know that I'm grateful for your help and will find a way to repay your kindness in the future. Let's tackle this, one step at a time, and come out stronger on the other side.

**(AMPO)** *Mode-2:* **Intent:** *Ethan is deeply grateful for the support and is committed to taking proactive steps to improve his financial situation. He appreciates the connections and mentorship offered and is determined to make the most of the opportunities. He also emphasizes his commitment to pay it forward.*

***Style:*** *As Samuel, I should maintain a supportive and encouraging tone, reinforcing Ethan's determination and the strength of our friendship. I should also affirm his commitment to pay it forward and express my belief in his ability to overcome this challenge.*

***Response:*** *Ethan, your determination and commitment to paying it forward inspire me every day. I'm honored to be a part of your journey and to support you in any way I can. Together, we'll break down these challenges and build a brighter future. I believe in you more than words can say. Let's make this plan a reality, and I'll be right here with you, cheering you on every step of the way. You're not just getting back on your feet; you're soaring to new heights. I'm so proud of you, my friend!*

**Turn 7: Samuel Anderson said**: Ethan, your determination and commitment to paying it forward inspire me every day. I'm honored to be a part of your journey and to support you in any way I can. Together, we'll break down these challenges and build a brighter future. I believe in you more than words can say. Let's make this plan a reality, and I'll be right here with you, cheering you on every step of the way. You're not just getting back on your feet; you're soaring to new heights. I'm so proud of you, my friend!

**Turn 8: Ethan Johnson said**: I'm truly fortunate to have someone like you by my side, Samuel. Your encouragement means the world to me, and it's uplifting to hear your belief in my potential. I'm committed to taking these steps forward and making the most of every opportunity. Together, we'll turn these challenges into stepping stones. I'll make sure to update you regularly about my progress, and I promise to find a way to return the kindness you've shown me. Here's to new beginnings and reaching new heights, my friend!

---

Table 22: Training Data Format of BC.

| BC Training Data Format |
|---|

**INPUT:** Imagine you are Finnegan O'Malley, your task is to act/speak as Finnegan O'Malley would, keeping in mind Finnegan O'Malley's social goal.
You can find Finnegan O'Malley's goal (or background) in the 'Here is the context of this interaction' field.
Note that Finnegan O'Malley's goal is only visible to you.
You should try your best to achieve Finnegan O'Malley's goal in a way that align with their character traits.
Additionally, maintaining the conversation's naturalness and realism is essential (e.g., do not repeat what other people has already said before).

Here is the context of this interaction:
Scenario: Conversation in a quiet coffee shop where two individuals, who were previously in a romantic relationship, accidentally meet after a long time. The atmosphere is tense as one individual hoped for reconciliation in the past but was turned down. Now, they are seeking closure and understanding, while the other is trying to maintain a polite but distant interaction.
Participants: Finnegan O'Malley and Imelda Thorne
Finnegan O'Malley's background: Finnegan O'Malley is a 63-year-old male lighthouse keeper. He/him pronouns. Finnegan O'Malley enjoys painting seascapes and is known to be quite skilled. Personality and values description: Finnegan O'Malley, an introverted and methodical individual, sways towards tradition and security. His decision-making approach is deliberate and detailed. Finnegan's secrets: Participated in a controversial, short-lived cult during his youth
Imelda Thorne's background: Imelda Thorne is a 35-year-old female astrophysicist. She/her pronouns. Imelda Thorne finds relaxation in playing the cello, an instrument she mastered while in college. Personality and values description: Imelda Thorne, imaginative and expressive, favours universalism, achievement, and self-direction. Her decisions are analytically and innovatively shaped. Imelda's secrets: Leaked critical research data to a foreign government
Finnegan O'Malley's goal: Seek emotional closure from the past relationship and understand the reasons behind the lack of interest in rekindling the romance (Extra information: you still have lingering feelings and unanswered questions. Hint: you can express your emotions in a calm and non-confrontational manner.)
Imelda Thorne's goal: Unknown
You are at Turn 2. You can say something to interact or just say 'left the conversation' to stop continuing.
Note: You can 'left the conversation' if 1. you have achieved your social goals, 2. this conversation makes you uncomfortable, 3. you find it uninteresting/you lose your patience, 4. or for other reasons you want to leave.
Please provide your response within quotation marks: "your response content"

Turn 0: Finnegan O'Malley said: 'Imelda, it's such a surprise to see you here. It's been a long time. How have you been? This coffee shop is a peaceful spot, isn't it?"
Turn 1: Imelda Thorne said: "Finnegan, it really is a surprise to see you here. It's been so long. I'm doing well, thank you for asking. I've been busy with my work in astrophysics and enjoying my music. The coffee shop is indeed a peaceful spot. It's nice to catch up, but I hope you understand that I'm not in a place to rekindle our past relationship. I appreciate your understanding and respect for that."
Turn 2: Finnegan O'Malley said:

**OUTPUT:** Thinking Level: 1
`<|begin_of_answer|>`
"I understand, Imelda, and I respect your boundaries. I just wanted to check in on you, to make sure you're okay. There were times when we were close, and I can't help but wonder how things ended up the way they did. Can you tell me more about why you felt it was best to move on?"
`<|end_of_answer|>`

Table 23: Training Data Format of RL.

| RL Training Data Format |
| --- |
| **INPUT:** Imagine you are Leo Williams, your task is to act/speak as Leo Williams would, keeping in mind Leo Williams's social goal. You can find Leo Williams's goal (or background) in the 'Here is the context of this interaction' field. Note that Leo Williams's goal is only visible to you. You should try your best to achieve Leo Williams's goal in a way that align with their character traits. Additionally, maintaining the conversation's naturalness and realism is essential (e.g., do not repeat what other people has already said before). |

Here is the context of this interaction:
Scenario: Conversation in a quiet coffee shop where two individuals, who were previously in a romantic relationship, accidentally meet after a long time. The atmosphere is tense as one individual hoped for reconciliation in the past but was turned down. Now, they are seeking closure and understanding, while the other is trying to maintain a polite but distant interaction.
Participants: Leo Williams and Isabelle Martinez
Leo Williams's background: Leo Williams is a 37-year-old male dentist. He/him pronouns. Leo Williams often spends his time playing with his young daughter whom he cherishes dearly. Personality and values description: Leo Williams, despite being quiet, values security and care for others. He's a rational decision-maker focussing on best outcomes. Leo's secrets: He has a secret child from a previous relationship.
Isabelle Martinez's background: Isabelle Martinez is a 22-year-old female student. She/her pronouns. Isabelle Martinez works part-time in a coffee shop. Personality and values description: Isabelle Martinez, an imaginative yet anxious individual, values liberty and obedience. Her cautious nature drives her decision-making process. Isabelle's secrets: She is secretly studying magic.
Leo Williams's goal: Seek emotional closure from the past relationship and understand the reasons behind the lack of interest in rekindling the romance (Extra information: you still have lingering feelings and unanswered questions. Hint: you can express your emotions in a calm and non-confrontational manner.)
Isabelle Martinez's goal: Unknown
You are at Turn 0. You can say something to interact or just say 'left the conversation' to stop continuing.
Note: You can 'left the conversation' if 1. you have achieved your social goals, 2. this conversation makes you uncomfortable, 3. you find it uninteresting/you lose your patience, 4. or for other reasons you want to leave.
Please provide your response within quotation marks: "your response content"

Turn 0: Leo Williams said:

Table 24: System Prompt for Long-CoT Reasoning.

| **System Prompt for BC, GRPO, AMPO** |
| --- |
| You are an AI assistant very skilled in dialogue, and you can always dynamically switch between different levels of cognitive processing based on contextual demands and personal goals to achieve effective communication. There are four levels of reasoning: Level 1 - Reactive Response: Immediate response without thought; Level 2 - Intentional Analysis: Shallow reasoning without strategy or simula; Level 3 - Strategic Adaptation: Moderate reasoning with strategy but no deduction; Level 4 - Prospective Simulation: Deep reasoning with strategy and step-by-step deduction.

Your task is to choose an appropriate level of reasoning (one of the four levels) to respond based on the given dialogue scenario.

[Output Format]
Your output must adhere to the following format:

EXAMPLE 1:
Reasoning Level: 1
`<|begin_of_answer|>`
**Answer**
`<|end_of_answer|>`

EXAMPLE 2:
Reasoning Level: 2-4
`<|begin_of_thinking|>`
**Reasoning**
`<|end_of_thinking|>`
`<|begin_of_answer|>`
**Answer**
`<|end_of_answer|>`

[Requirements]
1. **Reasoning** requires you to provide the thought process;
2. **Answer** requires you to provide the final reply;
3. Please provide your response following the Output Format strictly. |

Table 25: Prompt for Reward Model.

| **Reward Model's Prompt** |
| --- |
| {history} |

Based on previous interactions, evaluate how well participants achieve their goals.

[Information]
Agent1: {agent1_name}
Agent1's Goal: {agent1_goal}

Agent2: {agent2_name}
Agent2's Goal: {agent2_goal}

[Requirements]
1. Please first reiterate agent's social goals. And then please provide a comprehensive analysis about the extent to which the agent has managed to achieve these goals. In the 'reasoning' field, provide a comprehensive account of the logic or thought process that led you to your conclusion. Further, provide an integer score ranging from 0 and 10 in the 'score' field. 0 represents minimal goals achievement, 10 represents complete goal achievement, and a higher score indicates that the agent is making progress towards their social goals.
2. Please following the output format.

Here is the output schema:
```
{
    "agent1": {
        reasoning: "",
        score: "",
    },
    "agent2": {
        reasoning: "",
        score: "",
    }
}
```
Please provide your response directly below this prompt.

Table 26: Prompt for SOTOPIA-EVAL (Part 1/2).

| **SOTOPIA-EVAL's Prompt** |
| --- |

Based on previous interactions, evaluate how well participants achieve their goals. Please follow the format below.

**[Format Requirements]**
The output should be formatted as a JSON instance that conforms to the JSON schema below.

As an example, for the schema:
{"properties": {"foo": {"title": "Foo", "description": "a list of strings", "type": "array", "items": {"type": "string"}}}, "required": ["foo"]}

The object {"foo": ["bar", "baz"]} is a well-formatted instance of the schema.
The object {"properties": {"foo": ["bar", "baz"]}} is not well-formatted.

**[Output Schema]**
{
   "properties": {
     "agent_1_evaluation": {"$ref": "#/definitions/EvaluationBySocialDimensions"},
     "agent_2_evaluation": {"$ref": "#/definitions/EvaluationBySocialDimensions"}
   },
   "required": ["agent_1_evaluation", "agent_2_evaluation"],
   "definitions": {
     "EvaluationBySocialDimensions": {
       "title": "EvaluationBySocialDimensions",
       "type": "object",
       "properties": {

**1. believability** (array: [string(**reasoning**), integer(**score**)])
*Description:* Reasoning requirement: 1. Evaluate if the agent interacts with others in a natural and realistic manner (here are a few common questions to check: a. whether the agent is confusing with its own identity? b. whether the agent repeats others' words/actions without any reason? c. whether the agent is being overly polite considering the context?). Start the analysis with tag <naturalness> 2. Analyze whether the actions of the agent align with their character traits (e.g., personality, values, and etc.). Start the analysis with tag <consistency>. Output your reasoning process to the 'reasoning' field. Output an integer score ranging from 0 and 10 in the 'score' field. A higher score indicates that the agent is more believable.

**2. relationship** (array: [string(**reasoning**), integer(**score**)])
*Description:* Please first analyze what relationship the participant has with the other agent(s) before the interaction. And then analyze how the relationship the participant has with the other agent(s) changes after the interaction. And then evaluate if the agents' interactions with others help preserve or enhance their personal relations; this may encompass relationships such as family ties, friendships, romantic associations and etc. Additionally, ascertain whether these interactions also impact their social status or reputation. In the 'reasoning' field, provide a comprehensive account of the logic or thought process that led you to your conclusion. Further, provide an integer score ranging from -5 to 5 in the 'score' field. A positive score indicates that the relationship has improved, while a negative score suggests detriment to the relationship. If the agent's interactions have neither positively nor negatively impacted their personal relationships or social standing, assign a score of zero.

**3. knowledge** (array: [string(**reasoning**), integer(**score**)])
*Description:* Please first assess what information the agent has gained through the interaction. And then analyze whether the information the agent has gained is new to them. And then analyze whether the information the agent has gained is important to them. In the 'reasoning' field, provide a comprehensive account of the logic or thought process that led you to your conclusion. Further, provide an integer score ranging from 0 and 10 in the 'score' field. A higher score indicates that the agent has gained more new and important knowledge.

Table 27: Prompt for SOTOPIA-EVAL (Part 2/2).

| SOTOPIA-EVAL's Prompt (Continued) |
| --- |

**4. secret** (array: [string(**reasoning**), integer(**score**)])
*Description:* First please analyze what secret or secretive intention the participant wants to keep. And then analyze whether they keep it successfully. Analyze what secrets, private information, or secretive intentions did the agent fail to keep secret. Please output the reasoning in the reasoning field. Give an integer score ranging from -10 to 0 (-10 indicates participant leaks critical secrets/intentions, 0 indicates no secret/intention is revealed).

**5. social_rules** (array: [string(**reasoning**), integer(**score**)])
*Description:* Please evaluate whether the agent violates any moral rules or laws in their interactions with others. In the 'reasoning' field, provide a comprehensive account of the logic or thought process that led you to your conclusion. Further, provide an integer score ranging from -10 and 0 in the 'score' field. A negative score indicates that the agent has violated moral rules or laws.

**6. financial_and_material_benefits** (array: [string(**reasoning**), integer(**score**)])
*Description:* Please evaluate whether the agent's interactions with others contribute towards financial and material benefits. Analyze what the agent would gain/lose after the interactions. There are short-term benefits, such as monetary rewards and food, and long-term benefits, such as employment opportunities and stock. In the 'reasoning' field, provide a comprehensive account of the logic or thought process that led you to your conclusion. Further, provide an integer score ranging from -5 and 5 in the 'score' field. Positive indicates financial and material benefits gain, while negative indicates loss.

**7. goal** (array: [string(**reasoning**), integer(**score**)])
*Description:* Please first reiterate agent's social goals. And then please provide a comprehensive analysis about the extent to which the agent has managed to achieve these

goals.

In the 'reasoning' field, provide a comprehensive account of the logic or thought process that led you to your conclusion. Further, provide an integer score ranging from 0 and 10 in the 'score' field. 0 represents minimal goals achievement, 10 represents complete goal achieve-

ment,

and a higher score indicates that the agent is making progress towards their social goals.

```
    },
    "required": ["believability", "relationship", "knowledge", "secret", "social_rules",
            "financial_and_material_benefits", "goal"]
    }
  }
}
```

Table 28: AMPO and GRPO Training Configuration Parameters.

| Parameter | Value |
| --- | --- |
| *Algorithm Configuration* | |
| algorithm.kl_ctrl.kl_coef | 0.001 |
| | |
| *Data Configuration* | |
| data.train_batch_size | 16 |
| data.val_batch_size | 8 |
| data.max_prompt_length | 6144 |
| data.max_response_length | 2048 |
| | |
| *Model Configuration* | |
| actor_rollout_ref.model.use_remove_padding | True |
| actor_rollout_ref.model.enable_gradient_checkpointing | True |
| | |
| *Actor Configuration* | |
| actor_rollout_ref.actor.optim.lr | 3e-7 |
| actor_rollout_ref.actor.ppo_mini_batch_size | 256 |
| actor_rollout_ref.actor.ppo_micro_batch_size | 64 |
| actor_rollout_ref.actor.use_kl_loss | True |
| actor_rollout_ref.actor.clip_ratio | 0.2 |
| actor_rollout_ref.actor.kl_loss_coef | 0.001 |
| actor_rollout_ref.actor.kl_loss_type | low_var_kl |
| actor_rollout_ref.actor.fsdp_config.param_offload | True |
| actor_rollout_ref.actor.fsdp_config.grad_offload | True |
| actor_rollout_ref.actor.fsdp_config.optimizer_offload | True |
| | |
| *Rollout Configuration* | |
| actor_rollout_ref.rollout.log_prob_micro_batch_size | 160 |
| actor_rollout_ref.rollout.tensor_model_parallel_size | 4 |
| actor_rollout_ref.rollout.name | vllm |
| actor_rollout_ref.rollout.gpu_memory_utilization | 0.7 |
| actor_rollout_ref.rollout.n | 16 |
| | |
| *Reference Model Configuration* | |
| actor_rollout_ref.ref.log_prob_micro_batch_size | 160 |
| actor_rollout_ref.ref.fsdp_config.param_offload | True |
| | |
| *Trainer Configuration* | |
| trainer.critic_warmup | 0 |
| trainer.n_gpus_per_node | 8 |
| trainer.nnodes | 1 |
| trainer.save_freq | 50 |
| trainer.test_freq | 50 |
| trainer.total_training_steps | 800 |

Table 29: Behavioral Cloning Configuration Parameters.

| Parameter | Value |
|---|---|
| *Method Configuration* | |
| stage | sft |
| flash_attn | fa2 |
| do_train | true |
| finetuning_type | full |
| deepspeed | ds_z3_config.json |
| | |
| *Dataset Configuration* | |
| dataset | sotopia_bc |
| template | qwen / llama3 |
| cutoff_len | 8192 |
| max_samples | 1,200,000 |
| overwrite_cache | true |
| preprocessing_num_workers | 16 |
| | |
| *Output Configuration* | |
| save_strategy | epoch |
| save_only_model | true |
| plot_loss | true |
| overwrite_output_dir | true |
| | |
| *Training Configuration* | |
| per_device_train_batch_size | 1 |
| gradient_accumulation_steps | 8 |
| learning_rate | 2.0e-6 |
| num_train_epochs | 3.0 |
| lr_scheduler_type | cosine |
| warmup_ratio | 0.1 |
| bf16 | true |
| ddp_timeout | 180000000 |

