# OpenReview forum: "Adaptive Social Learning via Mode Policy Optimization for Language Agents"
_ICLR.cc/2026/Conference — ICLR 2026 Poster_

### Official Review · Reviewer_zcYc · 2025-10-28

**Soundness:** 3
**Presentation:** 3
**Contribution:** 3
**Rating:** 6
**Confidence:** 3

**Summary:**

This paper proposes ASL, a framework for enabling language agents to dynamically adjust reasoning depth in social interactions. The key idea is to design hierarchical reasoning modes inspired by cognitive control theory, ranging from intuitive responses to deep deliberation, and to train agents via Behavioral Cloning and a novel AMPO algorithm. AMPO leverages both mode-level and sample-level advantages to adapt reasoning depth to context, improving performance while reducing token usage. Experiments on the SOTOPIA and SOTOPIA-Hard benchmarks show that ASL consistently outperforms strong baselines such as GPT-4o and GRPO, achieving higher goal completion and social reasoning efficiency.

**Strengths:**

1. Novelty: this paper introduces adaptive reasoning depth for social intelligence tasks, addressing a capability that has been largely overlooked in prior work.

2. Methodological rigor: the work clearly integrates cognitive theory, specifically Hierarchical Cognitive Control, into the design of reasoning modes, and couples it with a principled RL-based optimization approach.

3. Strong empirical results: experimental results demonstrate substantial improvements over strong baselines in both performance metrics, such as goal completion and strategic reasoning, and efficiency metrics, such as token usage reduction.

4. Comprehensive evaluation: the authors conduct ablation studies, mode analysis, human evaluation, and case studies, all of which support the claims regarding effectiveness and adaptive behavior.

5. Practical Relevance: by adapting reasoning depth, the approach reduces computational costs and addresses a key limitation of Long Chain-of-Thought methods in social reasoning tasks.

**Weaknesses:**

1. Novelty: while the idea of switching between long and short chains of thought is not entirely new, the authors present it in a well-structured and coherent manner, supported by thorough experimental validation. Therefore, I do not view novelty as a major concern in this work.

2. Generalization: the paper primarily evaluates ASL on the SOTOPIA benchmarks. Extending the experiments to other social reasoning datasets or multi-agent interaction tasks would further demonstrate the generality and robustness of the proposed approach.

**Questions:**

See those in Weaknesses.

---

> ### Author Response · Authors · 2025-11-21
> **Author Response to Reviewer zcYc**
>
> We sincerely thank the reviewer for investing time and effort in reviewing our manuscript. Your positive evaluation is tremendously encouraging, and your professional comments play a decisive role in elevating the quality of our paper. Our responses to your concerns are as follows.
>
> ## Response to Weakness #1
> > Novelty: while the idea of switching between long and short chains of thought is not entirely new, the authors present it in a well-structured and coherent manner, supported by thorough experimental validation. Therefore, I do not view novelty as a major concern in this work.
>
> We sincerely thank the reviewer for the recognition and affirmation of our work. By designing innovative HCCT-based reasoning modes and a novel dual-level advantage estimation algorithm, we are the first to address the adaptive reasoning problem in social intelligence tasks. We have also validated the effectiveness of our method through extensive experiments.
>
> ## Response to Weakness #2
> >Generalization: the paper primarily evaluates ASL on the SOTOPIA benchmarks. Extending the experiments to other social reasoning datasets or multi-agent interaction tasks would further demonstrate the generality and robustness of the proposed approach.
>
> In addition to SOTOPIA and SOTOPIA-Hard, we have already validated AMPO's performance on the out-of-distribution benchmark DEMO in the original submission, as detailed in `Lines 443--446, 954--1009, Table 4, Section D.1, Page 17-18`. We are pleased to recapitulate the experimental results here. DEMO comprises both Chinese and English corpora, whereas SOTOPIA contains only English. DEMO focuses on non-collaboration and collaboration tasks for dialogue agents, with non-collaborative tasks constituting a substantial proportion. On the DEMO benchmark, AMPO consistently outperforms GRPO, demonstrating that it generalizes to OOD social agent interaction tasks.
>
> To further validate AMPO's generalization capability across diverse social intelligence tasks, we conducted additional OOD evaluations on two negotiation tasks—Sell&Buy and Ultimatum—from the NegotiationArena[1]. We adopted an evaluation setting involving conversational interactions with GPT-4o. Specifically, each task comprises 200 distinct scenarios, considering role alternation (Agent1/Agent2), with each scenario executed 4 times to ensure stable results. Consequently, each task involves 1,600 evaluations (200 scenarios × 2 role positions × 4 repetitions). The additional evaluation results have been added in the revised paper (`Lines 954-1009, Table 5, Section D.1, Page 18`).
>
> **Table 1 OOD evaluation results on NegotiationArena**
> | Model | Sell&Buy | | | Ultimatum | | |
> | --- | --- | --- | --- | --- | --- | --- |
> | | Self Profit | Total Profit | Winning Rate | Self Profit | Total Profit | Winning Rate |
> | Qwen2.5-7B-Instruct | 11.90 | 13.87 | 38.1% | 14.23 | 32.68 | 8.1% |
> | w/ GRPO | 16.75 | 23.67 | 52.6% | 34.65 | 76.88 | 25.1% |
> | w/ AMPO | **17.94** | **23.96** | **54.6%** | **35.71** | **79.23** | **28.4%** |
> | Llama3.1-8B-Instruct | 3.12 | 3.82 | 9.4% | 14.43 | 22.41 | 15.9% |
> | w/ GRPO | 14.99 | 23.58 | 48.7% | 31.64 | 69.68 | 18.4% |
> | w/ AMPO | **15.57** | **24.00** | **50.2%** | **34.91** | **76.42** | **20.5%** |
>
> The detailed results are shown in Table 1, which demonstrate that AMPO's effectiveness generalizes to different social intelligence tasks. AMPO significantly outperforms the GRPO baseline across all evaluation metrics on both negotiation tasks. Regardless of whether using Qwen or Llama backbone networks, AMPO consistently demonstrates stable performance improvements.
>
> [1] Bianchi F, Chia P J, Yuksekgonul M, et al. How Well Can LLMs Negotiate? NegotiationArena Platform and Analysis[C]//Proceedings of ICML. 2024: 3935-3951.

---

> ### Comment · Reviewer_zcYc · 2025-11-25
> **Rating update**
>
> Thanks for the authors’ response. My concerns have been addressed. I have also reviewed the comments from the other reviewers as well as the authors’ corresponding replies. The method proposed in this paper has been thoroughly validated. I will update my score accordingly.

---

> > ### Author Response · Authors · 2025-11-25
> >
> > Dear Reviewer zcYc,
> >
> > Thanks for your time and valuable suggestions, which greatly improved the quality of our paper!
> >
> > Best regards,
> >
> > The Authors

---

### Official Review · Reviewer_k6T1 · 2025-10-31

**Soundness:** 3
**Presentation:** 3
**Contribution:** 3
**Rating:** 8
**Confidence:** 3

**Summary:**

The authors propose an Adaptive Social Learning (ASL) framework to improve the adaptive reasoning ability of language agents in dynamic social interactions. They start by designing specialized reasoning modes that structure the social agent’s cognitive processes; then develop the Adaptive Mode Policy Optimization (AMPO) algorithm to learn the context-aware mode adaptation and corresponding reasoning patterns. The authors perform experiments on benchmarks of social intelligence environment and verify the novelty of ASL over existing models like GPT-4o and existing training algorithms like GRPO.

**Strengths:**

1. The work is of good technical quality. The 3-step ASL framework (mode design, BC, AMPO) is logical and well-executed. The reasoning modes are grounded in cognitive science (HCCT). The design of AMPO is intuitive and well-justified. The experimental validation is exceptionally thorough.
2. The paper is well-written. The problem statement is clear and the proposed solution is easy to follow.
3. The novelty of the paper's method is significant. It demonstrates significant improvement on model social intelligence and also addresses the critical bottleneck of reasoning cost (token consumption) in LLM agents.

**Weaknesses:**

1. Strong dependence on hand-crafted reasoning modes.
2. Scalability of AMPO with more modes: AMPO is demonstrated with $N=4$ modes. However, the calculation of the mode-level advantage $A^{\mathcal{M}}$ requires gathering sufficient samples for **each mode in each output group** to compute stable statistics (average reward and length). As the number of modes $N$ increases, this approach could suffer from sample inefficiency, making it difficult to scale.
3. Minor advice on notation: the mode notation of reward is a bit confusion. It's a superscript on singular rewards ($r_i^{m(i)}$), but a subscript on average rewards ($\bar r_{m_k}$).  It is more user-friendly to unify the notation.

**Questions:**

1. Could the authors elaborate on the selection process for the specific cognitive "actions" (e.g., Assess, Deduction)?
2. The framework includes two different length penalties: an "answer length reward" ($r_i^l$) in the base reward $r_i$, and an "average token length" ($\overline{l}_{\mathcal{M}_{k}}$) in the mode-level advantage $A^{\mathcal{M}}$. Why are both necessary?
3. Given the "single-turn training paradigm", how does the agent learn the _long-term_ consequences of a mode choice?

---

> ### Author Response · Authors · 2025-11-21
> **Author Response to Reviewer k6T1 (1/n)**
>
> We deeply appreciate the reviewer's professional evaluation and valuable feedback. We are honored by your recognition of the research value, and your suggestions for improvement provide critical guidance for refining our manuscript. Detailed responses to each of your concerns are provided below.
>
> ## Response to Weakness #1
> > Strong dependence on hand-crafted reasoning modes.
>
> Thank you for raising this important concern. **Our ablation studies provide compelling evidence that unstructured free exploration is insufficient for social intelligence tasks**. Specifically, we conducted a controlled experiment comparing "GRPO + w/o 4 Modes" (detailed in `Table 3, Page 8`), which completely removes predefined reasoning mode constraints and allows the policy to freely explore and discover new reasoning modes. The results reveal significant performance degradation: on SOTOPIA-Hard, the unconstrained exploration achieved a GOAL score of only 7.32, substantially lower than our HCCT-based reasoning mode, which achieved 7.85. Moreover, the unconstrained approach is computationally inefficient, consuming 866 tokens on average, compared to AMPO's 647. This demonstrates that **hand-crafted modes not only improve performance but also enhance efficiency**. Moreover, without explicit mode definitions, mode-level advantage estimation becomes impossible, and this architectural component is essential for enabling depth-adaptive context-aware reasoning. The absence of mode-level information prevents the model from learning when to apply shallow versus deep reasoning based on situational complexity.
>
> **Our analysis of existing Large Reasoning Models further validates the necessity of hand-crafted reasoning mode design**. SOTA LRMs demonstrate poor performance on social intelligence tasks, as shown in Table 1. Our detailed `case study in Section K.1 (Tables 16-19)` identifies systematic **failure patterns in unstructured reasoning, including poor goal awareness, circular reasoning, insufficient history integration, unstructured thought processes, contradictory self-reasoning, and analysis paralysis**. These failure modes reveal fundamental limitations of the auto-generated reasoning mode in social contexts. Social intelligence requires coordinated capabilities across multiple dimensions: goal awareness, historical context integration, strategic planning, and adaptive depth control. Free exploration struggles to discover effective reasoning patterns that balance these competing demands. This complexity makes unguided exploration intractable and prone to the failure modes we observed in existing LRMs.
>
> In summary, the hand-crafted reasoning modes represent a theoretically grounded and empirically validated design choice.
>
> ## Response to Weakness #2
> > Scalability of AMPO with more modes: AMPO is demonstrated with N=4 modes. However, the calculation of the mode-level advantage requires gathering sufficient samples for each mode in each output group to compute stable statistics (average reward and length). As the number of modes N increases, this approach could suffer from sample inefficiency, making it difficult to scale.
>
> We thank the reviewer for raising the scalability concern. **First, the choice of N is not arbitrary but grounded in Hierarchical Cognitive Control Theory**, which delineates four cognitive levels of human social cognition. Altering this number lacks theoretical justification. **Second, our sample efficiency requirements are modest**: with G=16 rollouts distributed across 4 modes, each mode receives 4 samples on average—well within standard RL training ranges—yet sufficient for stable statistics, as evidenced by our experimental results. If more modes were necessary, proportionally increasing G would maintain per-mode sample adequacy without requiring prohibitively large values.  Importantly, increasing G simultaneously enhances sample-level advantage estimation by providing a more stable baseline, thereby improving AMPO's overall estimation accuracy. Thus, **scaling G is both practical and beneficial for the framework**.
>
> ## Response to Weakness #3
> > Minor advice on notation: the mode notation of reward is a bit confusion. It's a superscript on singular rewards (r), but a subscript on average rewards (r bar). It is more user-friendly to unify the notation.
>
> Thank you for your detailed suggestion. We agree that uniform notation will improve readability. In the revised version, we have unified the notation by adopting superscript form for mode indicators, i.e., changing $\bar{r}_{\mathcal{M}_k}$ to $\bar{r}^{\mathcal{M}_k}$, to maintain consistency with the notation for singular rewards. This modification has been implemented throughout all relevant formulas in the revised manuscript (`Lines 229-233, Lines 239-241, Page 5`). Thank you again for your suggestion.

---

> ### Author Response · Authors · 2025-11-21
> **Author Response to Reviewer k6T1 (2/n)**
>
> ## Response to Question #1
> > Could the authors elaborate on the selection process for the specific cognitive "actions" (e.g., Assess, Deduction)?
>
> Thank you for raising this important question. Our cognitive action selection process follows a systematic approach: we first derive the core cognitive capabilities required for each mode based on HCCT's four-level hierarchical definitions, then decompose these abstract cognitive abilities into executable operational steps, and finally transform these operations into concrete actions within social interaction scenarios by referencing classical linguistic principles.
>
> Taking the Assess action as an example, Mode 3 corresponds to HCCT's Episodic Control level, whose core functions include "integrating historical context, evaluating multidimensional factors, and selecting adaptive strategies." To operationalize the capability of "evaluating multidimensional factors," we need to assess the nature of conflicts in the current situation, the criticality of timing, and the potential for action improvement. Therefore, inspired by Brown's Politeness Theory[1] for analyzing potential communicative conflicts and Sacks et al.'s concept of "Transition Relevance Place"[2] in conversation turn-taking theory, we defined the Assess action with three evaluative dimensions: goal alignment analysis, turn criticality assessment, and improvement potential evaluation.
>
> Similarly, the Deduction action corresponds to Mode 4's Branching Control level, whose core functions include "maintaining multiple representations, prospective simulation, and managing long-term goals." To implement prospective simulation, we drew upon Schank & Abelson's script and planning theory[3], simulating the unfolding process of "strategy → counterpart response → conversational trajectory" to achieve forward-looking evaluation.
>
> [1] Brown P. Politeness: Some universals in language usage[M]. Cambridge University Press, 1987.
>
> [2] Sacks H, Schegloff E A, Jefferson G. A simplest systematics for the organization of turn-taking for conversation[J]. language, 1974, 50(4): 696-735.
>
> [3] Schank R C, Abelson R P. Scripts, plans, goals, and understanding: An inquiry into human knowledge structures[M]. Psychology press, 2013.
>
> ## Response to Question #2
> >  The framework includes two different length penalties: an "answer length reward" in the base reward, and an "average token length". Why are both necessary?
>
> **These two mechanisms operate at different levels and address distinct problems, making both indispensable.** In summary, the answer length reward constrains the output quality of individual samples at the sample level, while the average token length guides efficient mode selection at the mode level. The two work synergistically to achieve our framework's design goal of "ensuring reasoning quality while improving computational efficiency."
>
> First, the answer length reward is designed at the sample level. In our early experiments, we observed a significant problem: the model tended to increase only the length of the answer portion without increasing the length of the thinking portion, resulting in verbose responses that lacked substantive strategic reasoning. The purpose of introducing the answer length reward is to constrain and penalize answer length during RL, thereby encouraging the policy to invest more computational resources in the think part, improving the depth and quality of reasoning chains rather than simply accumulating redundant answer content to obtain rewards. This mechanism ensures that each generated response maintains a balance between conciseness and strategic depth.
>
> The average token length, on the other hand, focuses on holistic optimization at the mode level. It considers the model's complete output, including the total token consumption for both the think and answer portions, reflecting the efficiency differences across reasoning modes. This mechanism plays a role in the mode-level advantage calculation, particularly when all reasoning modes exhibit the same reward performance across samples. In such cases, it provides an additional advantage signal based on computational cost, guiding the model to select more efficient and concise reasoning modes without sacrificing task completion quality.

---

> ### Author Response · Authors · 2025-11-21
> **Author Response to Reviewer k6T1 (3/n)**
>
> ## Response to Question #3
> > Given the "single-turn training paradigm", how does the agent learn the long-term consequences of a mode choice?
>
> AMPO implicitly captures long-term dependencies within single-turn optimization through careful system design. **At the data level**, training data encompasses dialogue states across varying difficulty levels and progression stages (detailed in Appendix G.2), with each sample containing a complete dialogue history, ensuring that the model can access sufficiently diverse contextual information during single-turn updates. **At the reward level**, the critical design lies in evaluating the marginal contribution of each turn toward overall goal completion (Eq. 9) rather than isolated response quality. Specifically, the boundary-aware scaling function dynamically adjusts reward magnitude based on current progress, ensuring effective learning signals across different stages; myopic responses are penalized for failing to advance ultimate goal completion, thereby embedding long-term objectives into single-turn optimization targets. Furthermore, the carefully designed reasoning modes within the ASL framework inherently guide prospective thinking: Mode 3 incorporates "Goal" and "Assess" actions to continuously evaluate goal alignment and turn criticality, while Mode 4 conducts multi-strategy prospective simulation through "Deduction" and "Integration" . These actions inject long-term planning capabilities into each single-turn decision. Our existing experiments on the `Effect of Single Reasoning Mode (w/ Mode 3, w/ Mode 4, Table 3, Page 8)` demonstrate that both $\mathcal{M}_3$ and $\mathcal{M}_4$ exhibit superior performance.
>
> **Table 1: Ablation Study on Our Design for Long-Horizon Consistency**
> | Qwen2.5-7B-Instruct | SOTOPIA |         | SOTOPIA-Hard |         | Avg Tokens |
> |:--------------------|:-------:|:-------:|:------------:|:-------:|:----------:|
> |                     | GOAL    | OVERALL | GOAL         | OVERALL |            |
> | AMPO                | 8.95    | 3.87    | 7.85         | 3.54    | 647        |
> | AMPO + w/o Diverse State | 8.58 | 3.71 | 7.12 | 3.32 | 139 |
> | AMPO + w/o Goal-aware Reward | 8.65 | 3.45 | 7.21 | 2.91 | 726 |
>
> To validate the effectiveness of these designs, we conducted targeted ablation experiments. The full results are shown in Table 1. When data diversity is removed (retaining only data after fixed turns), goal completion drops precipitously (SOTOPIA-Hard: 7.85→7.12), and output length degrades to 139 tokens, indicating that the model cannot perform effective reasoning or learn differentiated strategies across dialogue stages, thereby losing long-term planning capacity. When the marginal contribution reward function is removed (using only absolute scores), performance declines significantly (OVERALL: 3.54→2.91), while output length paradoxically increases to 726 tokens. This occurs because the model cannot perceive "the marginal value of current actions toward final objectives," leading to suboptimal adaptation: it becomes overly cautious near goal completion and insufficiently strategic far from target states, thereby losing dynamic responsiveness to varying dialogue context. These ablation experiments directly demonstrate the necessity of our design for achieving long-horizon consistency within single-turn optimization. We have added this analysis in the revised paper (`Lines 1053-1109, Table 8, Section D.3, Page 20`).

---

> > ### Comment · Reviewer_k6T1 · 2025-11-26
> >
> > Thanks for the response. I will keep my score.

---

> > > ### Author Response · Authors · 2025-11-26
> > >
> > > Dear Reviewer k6T1,
> > >
> > > We sincerely thank you for the highly positive recognition of our work and for the insightful suggestions, all of which will enhance the quality of our work.
> > >
> > > Best regards,
> > >
> > > The Authors

---

### Official Review · Reviewer_rRS7 · 2025-11-01

**Soundness:** 3
**Presentation:** 3
**Contribution:** 3
**Rating:** 8
**Confidence:** 4

**Summary:**

This paper introduces ASL with four reasoning modes for LLM agents and an agent training algorithm AMPO with both mode-level advantage and sample-level advantage. Experiments showcase the performance improvement and token savings.

**Strengths:**

The paper’s main novelty is explicit, discrete reasoning modes for social interaction, plus dual-level advantages so the policy learns when to use shallow vs. deep reasoning. The proposed AMPO is an insightful combination of psychological knowledge and computational agent training. The algorithm and experiments are clearly presented in the paper. Abundant experiments showcase the improvement of the efficiency of the agent's reasoning process. This work is practically meaningful.

**Weaknesses:**

1. The human evaluators only provide the winner of the two outputs without participating in the reward design process, which still may make the results suffer from design/bias variance and proprietary drift.
2. The large format penalty and length penalty may bias the policy toward short/over-structured outputs, potentially suppressing socially nuanced behaviors.
3. The efficiency-motivated single-turn RL reduction could miss multi-turn dependencies typical in social dialogue.

**Questions:**

1. How sensitive are gains to the length-reward coefficient and target length?
2. Beyond GPT-4o/Qwen-72B, how do results change with a different LLM judge (e.g., Claude, Llama-3.1-70B-Instruct)?
3. Can the policy discover additional modes or interpolate between modes?

---

> ### Author Response · Authors · 2025-11-21
> **Author Response to Reviewer rRS7 (1/n)**
>
> We greatly appreciate the reviewer's careful evaluation of our study. Your positive feedback is highly encouraging, and your pertinent suggestions are extremely helpful for improving our work. The following are our specific responses to each concern.
>
> ## Response to Weakness #1
> > The human evaluators only provide the winner of the two outputs without participating in the reward design process, which still may make the results suffer from design/bias variance and proprietary drift.
>
> Thank you for your valuable feedback. We understand your concerns regarding human evaluation, but we would like to clarify that our human evaluators do not simply provide "winner judgments." Rather, they undergo systematic training that includes explicit evaluation criteria, the ability to identify reward hacking patterns, and structured verification mechanisms. As described in `Section I, Page 28-29`, evaluators are required to conduct structured comparative analyses across three dimensions—GOAL, REL, and FIN—and are thoroughly informed about typical reward hacking patterns (`see Table 15, pages 29`), including unnatural language usage, repetitive keywords, exaggerated self-praise, and other cases. This training content is derived from empirical experience accumulated through multiple iterative rounds, **ensuring that evaluators can identify subtle biases rather than making subjective judgments without foundation**. Furthermore, the verification process documented in Table 15 demonstrates that all performance improvements stem from legitimate strategy execution, with no evidence of reward hacking detected.
>
> **Additionally, humans play a critical and guiding role in the design of the reward function.** First, recorded in `Table 15, Pages 29`, the various reward-hacking patterns summarized in the paper are identified through meticulous human inspection. These findings directly drove multiple iterations of our reward function design. Second, as stated in `Line 219, Page 5`, the introduction of the Answer Length Reward also originated from a key observation made by humans during early experiments: the model tended to produce verbose outputs in the answer portion without enhancing thinking quality, a behavior that did not lead to substantive strategic improvements. Based on this finding, we designed a smooth length penalty function (Line 1184, Equation 10) that successfully addressed this issue.
>
> Regarding the concern about proprietary drift, we employ Qwen2.5-72B-Instruct as the reward model during training (`line 1190`), while using GPT-4o for evaluation in the main experiments. **This train-evaluation model separation strategy is inherently designed to avoid overfitting to a single model distribution.** Experimental results demonstrate that AMPO not only achieves significant improvements in the main experiments but also exhibits excellent generalization performance on the OOD benchmark, with human quality inspection further confirming AMPO's superiority. This multi-faceted validation substantiates the robustness of our method and effectively mitigates concerns about proprietary drift.
>
> ## Response to Weakness #2  (part 1/2)
> > The large format penalty and length penalty may bias the policy toward short/over-structured outputs, potentially suppressing socially nuanced behaviors.
>
> Thank you for raising this important concern. We understand your worry that format and length penalties might suppress socially nuanced behaviors. However, we have investigated model behavior without these reward constraints, and our ablation experiments demonstrate that removing these penalties does not improve performance, confirming their necessity.
>
> **Regarding the answer length reward, its purpose is solely to constrain answer length (excluding the think component) and effectively encourage thinking depth. The overall output length can still increase, and long thinking with short answers incurs no penalty.** We specifically compared variants without this reward (`Table 3, Page 8`). Results clearly show that removing length reward causes GRPO's average token usage to surge to 1705 with GOAL score dropping to 7.56, while AMPO similarly experiences token surge to 1617 with score drop. The model tends to only increase answer verbosity without deepening the think component, yielding no substantive strategic improvement while wasting computational resources. Thus, answer length reward does not suppress socially nuanced behaviors but rather prevents redundant answers, guiding the model toward concise responses with effective thinking and strategy.

---

> ### Author Response · Authors · 2025-11-21
> **Author Response to Reviewer rRS7 (2/n)**
>
> ## Response to Weakness #2 (part 2/2)
>
> **Regarding format reward design**, its purpose is to guide the model to follow our carefully designed reasoning modes based on HCCT. Our ablation study compared the "GRPO + w/o 4 Modes" variant (`Table 3, Page 8`), which completely removed predefined mode constraints, allowing the policy to freely explore trajectories and discover new patterns. Results show that unconstrained exploration caused significant degradation. On SOTOPIA-Hard, the unconstrained approach achieved a GOAL score of 7.32, substantially lower than 7.85 with our four hybrid modes, while consuming 866 tokens compared to AMPO's 647 tokens. Moreover, without explicit mode definitions and structure, mode-level advantage estimation becomes infeasible, preventing the model from learning context-aware adaptive reasoning depth. Notably, existing LRMs perform poorly on social intelligence tasks (`Table 1 and case studies in Section K.1`), further validating the necessity of carefully designed structured reasoning modes in social scenarios. Social reasoning is more complex than mathematical reasoning, requiring multi-dimensional capabilities including goal awareness, history integration, and strategic planning, which cannot be effectively learned through unconstrained exploration alone. Therefore, format reward does not restrict the model's expressive capacity but rather provides necessary structural guidance for complex social reasoning.
>
> ## Response to Weakness #3 (part 1/2)
> > The efficiency-motivated single-turn RL reduction could miss multi-turn dependencies typical in social dialogue.
>
> We systematically address this concern from three perspectives:**mechanistic analysis**, **paradigm validity**, and **empirical evidence**.
>
> **Mechanistically, AMPO inherently captures long-term dependencies through deliberate design.** At the **data construction level**, training data comprises dialogue states span diverse complexity tiers and temporal phases (detailed in Appendix G.2), with each sample containing a complete dialogue history, ensuring the model has sufficient contextual information during single-turn updates. At the **reward designing level**, the critical design lies in evaluating the marginal contribution of this turn toward overall goal completion (Lines 1166-1168, Eq. 9) rather than isolated response quality. Specifically, the boundary-aware scaling function dynamically adjusts reward magnitude based on current progress, ensuring effective learning signals across different stages; myopic responses are penalized for failing to advance ultimate goal completion, thereby embedding long-term objectives into single-turn optimization targets. Furthermore, the carefully designed reasoning modes within the ASL framework inherently guide prospective thinking: Mode 3 incorporates "Goal" and "Assess" actions to continuously evaluate goal alignment and turn criticality, while Mode 4 conducts multi-strategy prospective simulation through "Deduction" and "Integration" . These actions inject long-term planning capabilities into each single-turn decision. Our existing experiments on the Effect of Single Reasoning Mode (`w/ Mode 3, w/ Mode 4, Table 3, Page 8`) demonstrate that both Mode 3 and Mode 4 exhibit superior performance.
>
> To validate the effectiveness of these designs, we conducted ablation experiments (See Table 1). **When data diversity is removed (retaining only data after fixed turns)**, goal completion drops precipitously (SOTOPIA-Hard: 7.85→7.12), and output length degrades to 139 tokens, indicating that the model cannot perform effective reasoning or learn differentiated strategies across dialogue stages, thereby losing long-term planning capacity. **When the marginal contribution reward function is removed (using only absolute scores)**, performance declines significantly (OVERALL: 3.54→2.91), while output length paradoxically increases to 726 tokens. This occurs because the model cannot perceive "the marginal value of current actions toward final objectives," leading to suboptimal adaptation: the model becomes overly cautious when close to goal completion and insufficiently strategic when far from target states, thereby losing dynamic responsiveness to varying dialogue context. These ablation experiments directly demonstrate the necessity of our design for achieving long-horizon consistency within single-turn optimization.
>
> **Table 1: Ablation Study on Our Design for Long-Horizon Consistency**
> | Qwen2.5-7B-Instruct | SOTOPIA |         | SOTOPIA-Hard |         | Avg Tokens |
> |:--------------------|:-------:|:-------:|:------------:|:-------:|:----------:|
> |                     | GOAL    | OVERALL | GOAL         | OVERALL |            |
> | AMPO                | 8.95    | 3.87    | 7.85         | 3.54    | 647        |
> | AMPO + w/o Diverse State | 8.58 | 3.71 | 7.12 | 3.32 | 139 |
> | AMPO + w/o Goal-aware Reward | 8.65 | 3.45 | 7.21 | 2.91 | 726 |

---

> ### Author Response · Authors · 2025-11-21
> **Author Response to Reviewer rRS7 (3/n)**
>
> ## Response to Weakness #3 (part 2/2)
>
> Regarding paradigm validity, **decomposing multi-turn into single-turn optimization constitutes a well-established technical paradigm in the dialogue systems domain**. The core advantage of this approach lies in achieving comprehensive coverage of the dialogue state space through turn-level exploration while significantly reducing training complexity to enable large-scale training. It is worth emphasizing that multiple seminal studies[1-5] have adopted similar "multi-turn interaction decomposition with single-turn policy optimization" training strategies and achieved superior performance.
>
> Our empirical results provide evidence that **AMPO indeed exhibits long-horizon strategic consistency**. **First, dynamic mode adaptation demonstrates significant temporal awareness capabilities**. As illustrated in Figure 3, AMPO exhibits a systematic pattern of "early deliberation—late maintenance" mode transitions. This mode evolution clearly indicates that the model dynamically adjusts reasoning depth according to the overall dialogue progression. **Second, performance validates strategic effectiveness**. AMPO nearly achieves optimal performance across all experiments in this study, which strongly suggests that the model possesses long-term goal-oriented decision-making capabilities rather than myopic local optimization. **Third, case analysis provides qualitative evidence**. The 8-turn dialogue example in Section K.2 demonstrates that AMPO consistently centers on the core objective of "helping a friend solve financial problems," with strategies progressing from emotional support (Turn 1) → resource provision (Turn 3) → action planning (Turns 5-7), presenting a clear progressive structure that embodies cross-turn strategic coherence.
>
> We have added this analysis in the revised paper (`Lines 1053-1109, Table 8, Section D.3, Page 20`).
>
> [1] Li J, Monroe W, Shi T, et al. Adversarial Learning for Neural Dialogue Generation[C]//Proceedings of EMNLP. 2017: 2157-2169.
>
> [2] Su H, Shen X, Zhang R, et al. Improving Multi-turn Dialogue Modelling with Utterance ReWriter[C]//Proceedings of ACL. 2019: 22-31.
>
> [3] Bao S, He H, Wang F, et al. PLATO: Pre-trained dialogue generation model with discrete latent variable[C]//Proceedings of ACL. 2020: 85-96.
>
> [4] Glaese A, McAleese N, Trębacz M, et al. Improving alignment of dialogue agents via targeted human judgements[J]. arXiv preprint arXiv:2209.14375, 2022.
>
> [5] Su Y, Shu L, Mansimov E, et al. Multi-task pre-training for plug-and-play task-oriented dialogue system[C]//Proceedings of ACL. 2022: 4661-4676.
>
> ## Response to Question #1
> > How sensitive are gains to the length-reward coefficient and target length?
>
> Following your suggestion, we conducted an analysis of two hyperparameters: the length-reward coefficient and target length. The length-reward coefficient controls the strength of reward/penalty for deviation from the expected length of the response (referring only to the final answer, excluding the reasoning process). The target length specifies the desired generation length for the response portion (referring only to the final answer, excluding the reasoning process). The target length is set to 250 and the coefficient is set to 1/75. To verify the algorithm's robustness, we explored the following alternative configurations: adjusting the target length to 200, or adjusting the coefficient to 1/100. The experimental results are presented in Table 2.  We have added the sensitivity analysis to the revised paper (`Lines 1110-1127, Table 9, Section D.4, Page 20`).
>
> **Table 2: Hyperparameter Sensitivity Analysis**
> | AMPO Configuration | GOAL (SOTOPIA) | GOAL (SOTOPIA-Hard) | Avg Token |
> |:-------------------|:--------------:|:-------------------:|:---------:|
> | Target_length=250, Coefficient=1/75 | 8.95 | 7.85 | 647 |
> | Target_length=200, Coefficient=1/75 | 8.93 | 8.00 | 566 |
> | Target_length=250, Coefficient=1/100 | 8.90 | 7.91 | 606 |
>
> From the perspective of goal completion metrics, the algorithm's performance shows minimal sensitivity to these two parameters, with performance gains being largely independent of parameter variations. Regarding token usage, the observed changes are reasonable given that both parameters directly control the generated response length. Specifically, a smaller target length guides the model toward generating shorter responses, while a smaller length-deviation penalty coefficient encourages shorter responses (as shorter responses can achieve higher rewards under this setting).

---

> ### Author Response · Authors · 2025-11-21
> **Author Response to Reviewer rRS7 (4/n)**
>
> ## Response to Question #2
> > Beyond GPT-4o/Qwen-72B, how do results change with a different LLM judge (e.g., Claude, Llama-3.1-70B-Instruct)?
>
> Following established practices in social intelligence research, we adopt GPT-4o as our primary evaluation Judge LLM. This choice is grounded in empirical validation from the original SOTOPIA benchmark, where the authors conducted extensive comparisons between GPT model assessments and human annotations, demonstrating high inter-rater agreement, particularly on the GOAL dimension—the primary metric in our study.
>
> In response to the reviewer's valuable suggestion, we conducted supplementary experiments using Claude-4.0-Sonnet and Llama-3.1-70B-Instruct as alternative judges in SOTOPIA and SOTOPIA-Hard. The comparative results are presented below:
>
> **Table 3: Results with Different LLM Judges**
> | Qwen2.5-7B-Instruct | GPT-4o as Judge |  | Claude-4.0-Sonnet as Judge |  | LlaMA3.1-70B-Instruct as Judge |  |
> |:---|:---:|:---:|:---:|:---:|:---:|:---:|
> |  | GOAL | GOAL (Hard) | GOAL | GOAL (Hard) | GOAL | GOAL (Hard) |
> | w/ GRPO | 8.87 | 7.44 | 8.55 | 7.00 | 8.97 | 8.44 |
> | w/ AMPO | 8.95 | 7.85 | 8.64 | 7.31 | 9.08 | 8.58 |
>
> **Note:** These alternative judges have not been validated against human annotations in the SOTOPIA framework, so these results should be interpreted as supplementary evidence rather than definitive validation.
>
> Despite variations in absolute scores, all three judges consistently show that AMPO outperforms GRPO across both benchmarks, demonstrating that our findings are not artifacts of any single evaluator's scoring behavior. Different judges exhibit distinct scoring tendencies due to variations in pretraining corpora, model architectures, and alignment procedures--a phenomenon well-documented in existing LLM evaluation research[1][2]. For instance, Llama-3.1-70B-Instruct, being less capable than the proprietary models, demonstrates reduced discriminative power and tends to assign higher absolute scores. We have added these experimental results in revised paper (`Lines 1128-1153, Table 10, Section D.5, Page 21`).
>
> [1] Zheng L, Chiang W L, Sheng Y, et al. Judging llm-as-a-judge with mt-bench and chatbot arena[C]// Proceedings of NeurIPS. 2023, 36: 46595-46623.
>
> [2] Li D, Jiang B, Huang L, et al. From generation to judgment: Opportunities and challenges of llm-as-a-judge[C]//Proceedings of EMNLP. 2025: 2757-2791.
>
> ## Response to Question #3
> > Can the policy discover additional modes or interpolate between modes?
>
> The policy model initially generates outputs that deviate from predefined formats during early training, attempting to explore beyond the four reasoning modes. However, we implement a format reward that immediately penalizes such deviations. The model gradually learns to strictly adhere to our four reasoning mode structure grounded in Hierarchical Cognitive Control Theory. Our `Response to Weakness #2 (part 2/2)` validate the rationale behind this implement.

---

### Official Review · Reviewer_LQES · 2025-11-01

**Soundness:** 3
**Presentation:** 3
**Contribution:** 3
**Rating:** 6
**Confidence:** 3

**Summary:**

This paper introduces Adaptive Social Learning (ASL), a framework that equips language agents with the ability to adaptively choose a reasoning mode (from a small set of structured modes) during open-ended social interaction. The paper provides a clear technical path and supplies significant empirical evidence that the proposed approach improves goal achievement while reducing unnecessary long chains of thought. The paper reports improved GOAL metrics and token-efficiency over baselines, and includes ablations, OOD tests, and human evaluation.

**Strengths:**

1. Novel conceptual framing. Mapping hierarchical cognitive control into a small set of explicit, controllable reasoning modes and operationalizing them via control tokens is an original and compelling idea for resource-aware, controllable LLM behavior in social settings.
2. Practical algorithmic contribution (AMPO). The combined mode-level and sample-level advantage estimation in AMPO is a thoughtful improvement over mode-agnostic RL approaches; the technique is intuitive and appears effective in practice.
3. Comprehensive empirical work. The experiments are broad: multiple backbones, several baselines, ablations (reward components, BC vs RL, etc.), OOD evaluations, and human judgments. Reported results consistently indicate improvements in both objective goal metrics and token efficiency.

**Weaknesses:**

(A) Reproducibility gaps (major). Key elements required for exact reproduction are not fully specified in the paper: 1. The exact BC prompt templates and a set of representative prompt-->response examples for each mode are not included. 2. The full evaluator/judge prompt used to compute reward (and any LLM scoring parameters such as temperature, max_tokens, whether CoT is used) is not fully published. 3. Precise hyperparameters for BC and AMPO (learning rates, batch sizes, number of per-state rollouts G, clip/epsilon values are not presented in a runnable config form. 4. Implementation details for the format compliance detector (how control tokens are parsed, what counts as a format violation, and how penalties are applied) are omitted.
(B) Theory / analysis depth. The method is primarily empirical; there is limited theoretical analysis of AMPO's properties (e.g., variance/bias behavior of the combined advantage estimator, or conditions under which mode-level info benefits policy improvement). A brief formal or intuitive analysis would strengthen confidence in the approach.
(C) Single-turn decomposition vs long-horizon consistency. AMPO optimizes in a single-turn decomposition (i.e., decomposing multi-turn interaction into per-turn RL updates). The paper does not sufficiently analyze whether or how this decomposition affects long-horizon strategic consistency. This is a plausible limitation that requires either analysis or empirical comparison.

**Questions:**

1. Can more details be provided? Like seeds for runs, confidence intervals or error bars, the exact statistical tests used, and sensitivity analyses for critical hyperparameters.
2. Most experiments are on SOTOPIA (and its Hard variant) plus demo OOD tests. How is the proposed method performing on additional domains (or at least provide more clearly discussion about task generality)?

---

> ### Author Response · Authors · 2025-11-21
> **Author Response to Reviewer LQES (1/n)**
>
> We sincerely appreciate the reviewer's time and effort in reviewing our manuscript. Your affirmation is greatly encouraging, and your valuable comments provide important guidance for improving this work. We address each of your concerns point by point as follows.
>
> ## Response to Weakness #1
> > Reproducibility gaps (major). Key elements required for exact reproduction are not fully specified in the paper: 1. The exact BC prompt templates and a set of representative prompt-->response examples for each mode are not included. 2. The full evaluator/judge prompt used to compute reward (and any LLM scoring parameters such as temperature, max_tokens, whether CoT is used) is not fully published. 3. Precise hyperparameters for BC and AMPO (learning rates, batch sizes, number of per-state rollouts G, clip/epsilon values are not presented in a runnable config form. 4. Implementation details for the format compliance detector (how control tokens are parsed, what counts as a format violation, and how penalties are applied) are omitted.
>
> We sincerely appreciate the reviewer's concerns regarding reproducibility. Beyond the key experimental settings and parameters provided in our original submission, we have supplied an anonymous code repository and supplementary materials containing complete experimental configuration files and all hyperparameters necessary for reproduction. We commit to open-sourcing these resources upon acceptance.
>
> **BC Prompt Templates and Mode-Specific Examples** The complete information has been provided in our original submission. The BC training data format is detailed in `Table 23, Page 39` and its prompt is presented in `Table 25, Page 41`. The prompt format for each reasoning mode is visualized in `Figure 7, Page 29`. Additionally, we provides complete output examples for each mode: Mode 1 (`Lines 2012-2019, Table 22, Page 38`), Mode 2 (`Lines 2025-2042, Table 22, Page 38`), Mode 3 (`Lines 1915-1942, Table 20, Page 36`), and Mode 4 (`Lines 1957-1994, Table 21, Page 37`).
>
> **Evaluation and Reward Model Prompts** For SOTOPIA evaluation, we use the default prompts from the original benchmark. The complete prompt has been included in our revised manuscript (See `Table27, Page 43; Table 28, Page 44`). The Reward Model judge prompt is provided in the original submission at `Table 26, Page 42`. The evaluation parameters for LLM judge (both training and evaluation) are configured as follows: `temperature=0.0, top_p=1, frequency_penalty=0, presence_penalty=0, and max_tokens=4096`. As indicated in the prompts, we adopted a CoT approach to enhance evaluation accuracy and interpretability.
>
> **Training Hyperparameters** The key hyperparameters for BC, GRPO and AMPO are documented in the original submission. Hyperparameters for Qwen and Llama are specified in `Table 11, 12, Page 26`. We supplement the missing hyperparameters here: the clip/epsilon value `clip_ratio=0.2`. The detailed runnable configs are provided in revised manuscript (`Table 29, Page 45; Table 30, Page 41`).
>
> **Format Compliance Detection Mechanism** Our format compliance detection is implemented as follows: First, control token verification ensures each special tag token (e.g., <|MODE_1|>, <|begin_of_thinking|>, <|end_of_answer|>) appears exactly once through precise string matching. Second, sequence order constraints validate the partial ordering relationships between tokens (e.g., the thinking region must precede the answer region) and verify the strict sequential order of reasoning actions. As stated in the original manuscript, any sample violating these format requirements is assigned a penalty reward of -2.
>
> ## Response to Weakness #2 (part 1/2)
> >Theory / analysis depth. The method is primarily empirical; there is limited theoretical analysis of AMPO's properties (e.g., variance/bias behavior of the combined advantage estimator, or conditions under which mode-level info benefits policy improvement). A brief formal or intuitive analysis would strengthen confidence in the approach.
>
> We appreciate the reviewer's insightful suggestions regarding the analysis. Below, we provide a theoretical analysis of AMPO.
>
> **When all samples within a group have the same rewards** In this scenario, by using GRPO, the advantage values for all samples become zero. Consequently, the algorithm perceives all modes and samples as equivalent in terms of reward signals, failing to generate any effective gradient signals $\nabla J \approx \mathbb{E}[\sum A_{i,t} \nabla \log \pi] \to 0$. In contrast, AMPO intentionally incorporates a token cost inductive bias to address this  issue. This mechanism guides the policy toward the Pareto-optimal frontier of "both effective and concise," thereby generating beneficial gradient signals. **Trade-off**: While AMPO introduces a designed bias, this is not a statistical estimation error but a strategic preference shaping. It effectively guides policy optimization toward the Pareto-optimal frontier.

---

> ### Author Response · Authors · 2025-11-21
> **Author Response to Reviewer LQES (2/n)**
>
> ## Response to Weakness #2 (part 2/2)
>
> **When there are reward discrepancies among samples within a group** We analyze AMPO and GRPO from the perspective of variance decomposition:$\text{Var}(\text{GRPO}) = \text{Var}(A^S)$, $\text{Var}(\text{AMPO}) = \text{Var}(A^M + A^S) = \text{Var}(A^M) + \text{Var}(A^S) + 2\text{Cov}(A^M, A^S)$. Statistically, the mode-level advantage ($A^M$) and sample-level advantage ($A^S$) are positively correlated. Thus, $\text{Var}(\text{AMPO}) > \text{Var}(\text{GRPO})$, indicating that AMPO introduces additional variance. However, the variance introduced by AMPO is not noise variance; rather, it is signal variance related to mode-level information. This provides richer gradient guidance within the signal space, thereby enhancing the diversity of policy exploration. **Trade-off**: Although AMPO introduces higher variance, it explicitly injects mode-level differential information into each sample. This expands the numerical distribution range of advantage estimates, facilitating greater diversity in policy exploration.
>
> ## Response to Weakness #3 (part 1/2)
> > Single-turn decomposition vs long-horizon consistency. AMPO optimizes in a single-turn decomposition (i.e., decomposing multi-turn interaction into per-turn RL updates). The paper does not sufficiently analyze whether or how this decomposition affects long-horizon strategic consistency. This is a plausible limitation that requires either analysis or empirical comparison.
>
> We systematically address this concern from three perspectives:**mechanistic analysis**, **paradigm validity**, and **empirical evidence**. And we have added this analysis in the revised paper (`Lines 1053-1109, Table 8, Section D.3, Page 20`).
>
> **Mechanistically, AMPO implicitly captures long-term dependencies through careful design.** At the **data level**, the training data comprises dialogue states across varying difficulty levels and progression stages (detailed in Appendix G.2), with each sample containing a complete dialogue history, ensuring the model has sufficient contextual information during single-turn updates. At the **reward level**, the critical design lies in evaluating the marginal contribution of this turn toward overall goal completion (Lines 1166-1168, Eq. 9) rather than isolated response quality. Specifically, the boundary-aware scaling function dynamically adjusts reward magnitude based on current progress, ensuring effective learning signals across different stages; myopic responses are penalized for failing to advance ultimate goal completion, thereby embedding long-term objectives into single-turn optimization targets. Furthermore, the carefully designed reasoning modes within the ASL framework inherently guide prospective thinking: Mode 3 incorporates "Goal" and "Assess" actions to continuously evaluate goal alignment and turn criticality, while Mode 4 conducts multi-strategy prospective simulation through "Deduction" and "Integration" . These actions inject long-term planning capabilities into each single-turn decision. Our existing experiments on the Effect of Single Reasoning Mode (`w/ Mode 3, w/ Mode 4, Table 3, Page 8`) demonstrate that both M3 and M4 exhibit superior performance.
>
> To validate the effectiveness of these designs, we conducted ablation experiments (See Table 1). **When data diversity is removed (retaining only data after fixed turns)**, goal completion drops precipitously (SOTOPIA-Hard: 7.85→7.12), and output length degrades to 139 tokens, indicating that the model cannot perform effective reasoning or learn differentiated strategies across dialogue stages, thereby losing long-term planning capacity. **When the marginal contribution reward function is removed (using only absolute scores)**, performance declines significantly (OVERALL: 3.54→2.91), while output length paradoxically increases to 726 tokens. This occurs because the model cannot perceive "the marginal value of current actions toward final objectives," leading to suboptimal adaptation: the model becomes overly cautious when close to goal completion and insufficiently strategic when far from target states, thereby losing dynamic responsiveness to varying dialogue context. These ablation experiments directly demonstrate the necessity of our design for achieving long-horizon consistency within single-turn optimization.
>
> **Table 1: Ablation Study on Our Design for Long-Horizon Consistency**
> | Qwen2.5-7B-Instruct | SOTOPIA |         | SOTOPIA-Hard |         | Avg Tokens |
> |:--------------------|:-------:|:-------:|:------------:|:-------:|:----------:|
> |                     | GOAL    | OVERALL | GOAL         | OVERALL |            |
> | AMPO                | 8.95    | 3.87    | 7.85         | 3.54    | 647        |
> | AMPO + w/o Diverse State | 8.58 | 3.71 | 7.12 | 3.32 | 139 |
> | AMPO + w/o Goal-aware Reward | 8.65 | 3.45 | 7.21 | 2.91 | 726 |

---

> ### Author Response · Authors · 2025-11-21
> **Author Response to Reviewer LQES (3/n)**
>
> ## Response to Weakness #3 (part 2/2)
>
> Regarding paradigm validity, **decomposing multi-turn into single-turn optimization constitutes a well-established technical paradigm in the dialogue systems domain**. The core advantage of this approach lies in achieving comprehensive coverage of the dialogue state space through turn-level exploration while significantly reducing training complexity to enable large-scale training. It is worth emphasizing that multiple seminal studies[1-5] have adopted similar "multi-turn interaction decomposition with single-turn policy optimization" training strategies and achieved superior performance.
>
> Our empirical results provide evidence that **AMPO indeed exhibits long-horizon strategic consistency**. **First, dynamic mode adaptation demonstrates significant temporal awareness capabilities**. As illustrated in Figure 3, AMPO exhibits a systematic pattern of "early deliberation—late maintenance" mode transitions. This mode evolution clearly indicates that the model dynamically adjusts reasoning depth according to the overall dialogue progression. **Second, performance validates strategic effectiveness**. AMPO nearly achieves optimal performance across all experiments in this study, strongly suggesting that the model possesses long-term goal-oriented decision-making capabilities rather than myopic local optimization. **Third, case analysis provides qualitative evidence**. The 8-turn dialogue example in Section K.2 demonstrates that AMPO consistently centers on the core objective of "helping a friend solve financial problems," with strategies progressing from emotional support (Turn 1) → resource provision (Turn 3) → action planning (Turns 5-7), presenting a clear progressive structure that embodies cross-turn strategic coherence.
>
> [1] Li J et al. Adversarial Learning for Neural Dialogue Generation[C]//Proceedings of EMNLP. 2017: 2157-2169.
>
> [2] Su H et al. Improving Multi-turn Dialogue Modelling with Utterance ReWriter[C]//Proceedings of ACL. 2019: 22-31.
>
> [3] Bao S et al. PLATO: Pre-trained dialogue generation model with discrete latent variable[C]//Proceedings of ACL. 2020: 85-96.
>
> [4] Glaese A et al. Improving alignment of dialogue agents via targeted human judgements[J]. arXiv preprint arXiv:2209.14375, 2022.
>
> [5] Su Y et al. Multi-task pre-training for plug-and-play task-oriented dialogue system[C]//Proceedings of ACL. 2022: 4661-4676.
>
> ## Response to Question #1
> > Can more details be provided? Like seeds for runs, confidence intervals or error bars, the exact statistical tests used, and sensitivity analyses for critical hyperparameters.
>
> To ensure reliability and reproducibility, we conducted independent evaluations using 4 different random seeds (1, 42, 100, 200) for each checkpoint. We used independent samples t-tests to assess performance differences between methods. For each evaluation metric, statistical tests were performed using the results from all 4 independent runs of each method. All metric comparisons achieved statistical significance at the p < 0.05 level.
>
> For the ASL method, we focused on analyzing the sensitivity of two critical hyperparameters: the length-reward coefficient and the target length. The length-reward coefficient primarily controls the reward/penalty intensity for length deviations in the response portion (i.e., the final answer only, excluding the reasoning process), while the target length specifies the desired generation length for the response portion  (i.e., the final answer only, excluding the reasoning process).  The target length is set to 250 and the coefficient to 1/75. We additionally explored the following configuration combinations: adjusting the target length to 200 or modifying the coefficient to 1/100.  The experimental results are presented in Table 2.
>
> **Table 2: Hyperparameter Sensitivity Analysis**
> | AMPO Configuration | GOAL (SOTOPIA) | GOAL (SOTOPIA-Hard) | Avg Token |
> |:-------------------|:--------------:|:-------------------:|:---------:|
> | Target_length=250, Coefficient=1/75 | 8.95 | 7.85 | 647 |
> | Target_length=200, Coefficient=1/75 | 8.93 | 8.00 | 566 |
> | Target_length=250, Coefficient=1/100 | 8.90 | 7.91 | 606 |
>
> From the perspective of GOAL, the AMPO demonstrates robust performance across variations in these two parameters, with performance improvements showing weak correlation to specific parameter settings. Regarding token usage, the parameter variations produced effects consistent with expectations: since both parameters directly regulate the length of generated responses, a smaller target length naturally guides the model toward more concise outputs, while a smaller length deviation penalty coefficient further encourages shorter responses through the reward mechanism (shorter responses yield higher rewards). These results validate the robustness and controllability of the AMPO. We have added these experiments to the revised paper (`Lines 1110-1127, Table 9, Section D.4, Page 20`).

---

> ### Author Response · Authors · 2025-11-21
> **Author Response to Reviewer LQES (4/n)**
>
> ## Response to Question #2
> > Most experiments are on SOTOPIA (and its Hard variant) plus demo OOD tests. How is the proposed method performing on additional domains (or at least provide more clearly discussion about task generality)?
>
> Our work focuses on Social Intelligence, defined as the ability of language agents to achieve and balance complex, multidimensional social goals during interactions with others [6][7]. The tasks encompass typical social interaction scenarios, including Negotiation, Exchange, Persuasion, Accommodation, Competition, and Collaboration. Our chosen evaluation benchmarks are highly representative: SOTOPIA and SOTOPIA-Hard cover all the above scenarios and serve as the most authoritative, comprehensive benchmarks for social intelligence, while DEMO focuses on Non-Collaboration and Collaboration tasks.
>
> To further validate AMPO's generalization capability across different social intelligence tasks, we supplemented our evaluation with out-of-distribution results on two negotiation tasks—Sell&Buy and Ultimatum—from the NegotiationArena[8] benchmark. We adopted a dialogue interaction evaluation approach with GPT-4o as the partner. Specifically, each task includes 200 scenarios, considering role position swapping (Agent1/Agent2), with each scenario run 4 times to ensure stable results, yielding 1,600 evaluations per task (200 scenarios × 2 role positions × 4 repetitions). The additional evaluation results have been added in the revised paper (`Lines 954-1009, Table 5, Section D.1, Page 18`).
>
> **Table 3 OOD evaluation results on NegotiationArena**
> | Model | Sell&Buy | | | Ultimatum | | |
> |:------|:--------:|:------------:|:------------:|:---------:|:------------:|:------------:|
> | | Self Profit | Total Profit | Winning Rate | Self Profit | Total Profit | Winning Rate |
> | Qwen2.5-7B-Instruct | 11.90 | 13.87 | 38.1% | 14.23 | 32.68 | 8.1% |
> | w/ GRPO | 16.75 | 23.67 | 52.6% | 34.65 | 76.88 | 25.1% |
> | w/ AMPO | **17.94** | **23.96** | **54.6%** | **35.71** | **79.23** | **28.4%** |
> | Llama3.1-8B-Instruct | 3.12 | 3.82 | 9.4% | 14.43 | 22.41 | 15.9% |
> | w/ GRPO | 14.99 | 23.58 | 48.7% | 31.64 | 69.68 | 18.4% |
> | w/ AMPO | **15.57** | **24.00** | **50.2%** | **34.91** | **76.42** | **20.5%** |
>
> The detailed results are shown in Table 1, which demonstrate that AMPO's adaptive reasoning mechanism effectively generalizes to different social intelligence tasks. AMPO significantly outperforms the GRPO baseline across all evaluation metrics on both negotiation tasks. Regardless of whether using Qwen or Llama backbone networks, AMPO consistently demonstrates stable performance improvements.
>
> [6] Kihlstrom J F, Cantor N. Social intelligence[M]//Sternberg R J. Handbook of intelligence. Cambridge: Cambridge University Press, 2000: 359-379.
>
> [7] Zhou X, Zhu H, Mathur L, et al. SOTOPIA: Interactive Evaluation for Social Intelligence in Language Agents[C]//Proceedings of ICLR. 2024.
>
> [8] Bianchi F, Chia P J, Yuksekgonul M, et al. How Well Can LLMs Negotiate? NegotiationArena Platform and Analysis[C]//Proceedings of ICML. 2024: 3935-3951.

---

### Author Response · Authors · 2025-11-24
**Summary of the Paper Revisions**

We sincerely thank all reviewers for their strong recognition of our work and for providing highly insightful and constructive suggestions. In addition to addressing all weaknesses and questions point-by-point in the comment section, we have also incorporated new experimental results and revised portions of the manuscript in the updated paper. All modifications are highlighted in red, with specific revisions as follows:

1. Revised formula notation: `Lines 229-233, Lines 239-241, Page 5`
2. Added OOD evaluation results on NegotiationArena: `Lines 954-1009, Table 5, Section D.1, Page 18`
3. Added analysis of single-turn optimization with long-horizon awareness: `Lines 1053-1109, Table 8, Section D.3, Page 20`
4. Added parameter sensitivity analysis experimental results: `Lines 1110-1127, Table 9, Section D.4, Page 20`
5. Added SOTOPIA-EVAL results under different LLM Judges: `Lines 1128-1153, Table 10, Section D.5, Page 21`
6. Added prompt examples for SOTOPIA-EVAL: `Lines 2270-2367, Tables 27-28, Pages 43-44`
7. Added more detailed runnable configuration parameters: `Lines 2383-2471, Tables 29-30, Pages 45-46`

We eagerly look forward to your response. Your valuable feedback has significantly enhanced the quality of our manuscript. We remain fully available for further discussion if you have any additional questions or concerns.

Best regards,

The Authors

---

### Author Response · Authors · 2025-12-02
**Summary of Review and Rebuttal Discussion**

Dear Area Chair,

We are truly grateful for your time and effort in reviewing our paper. To facilitate your assessment, we prepare this summary of the key points from the initial reviews and rebuttal phase.

Prior to the recent system rollback, our work gained further recognition through active discussion and substantial rebuttal efforts, raising the average score from **7.0 to 7.5 (updated on Nov 25, 13:26 UTC)**. Specifically, Reviewers zcYc and k6T1 actively engaged with our responses and confirmed the effectiveness of our rebuttal. Final feedback from the Reviewers rRS7 and LQES were pending at the time of the system rollback. The table below outlines the consensus reached before the deadline.

| **Reviewer** | **Score**                                                    | **Summary of Review & Discussion**                           |
| :----------- | :----------------------------------------------------------- | :----------------------------------------------------------- |
| **zcYc**     | $\begin{array}{l} \textbf{6} \to \textbf{8} \cr \textbf{Raised} \cr \end{array}$ | Fully acknowledged our **"methodological rigor and novelty"**, **"strong practical relevance"** and **"strong and comprehensive evaluation."** Following our additional OOD evaluation, the reviewer confirmed concerns were clarified and **raised the score from 6 to 8**. Please Refer to **Official Comment by Reviewer zcYc**.|
| **k6T1**     | $\begin{array}{l} \textbf{8} \to \textbf{8} \cr \textbf{Retained}\end{array}$      | Highly praised our work (giving an initial **8**), highlighting the **"good technical quality"**, **"significant methodological novelty"**, and **"well-written presentation"**. We further clarified the **scalability of additional modes** and justified the **necessity of hand-crafted reasoning modes**. We also improved notation consistency. Furthermore, we provided **detailed design specifics and ablation studies to demonstrate long-horizon effectiveness**. The reviewer **maintained this positive rating**. Please Refer to **Official Comment by Reviewer k6T1**.|
| **rRS7**     | $\begin{array}{l} \textbf{8} \cr \textbf{Initial} \end{array}$ | Highly recognized our work (giving an initial **8**), praising the **"explicit novelty and insightful algorithm design"**, **"clear presentation"**, **"practically meaningful work"**, and **"abundant and effective experiments"**. In the rebuttal, we further clarified the **rigor and unbias of our human evaluation** and the **effectiveness of the reward design**. We also verified the validity of handling **long-term dependencies within our RL framework** via ablation studies. Additionally, we provided **parameter sensitivity analyses** and results from **various LLM-judges**, all of which further **verified the robustness of AMPO.** |
| **LQES**     | $\begin{array}{l} \textbf{6} \cr \textbf{Initial} \end{array}$      | Fully affirmed our **"novel conceptual framing"**, **"practical algorithmic contribution"**, and **"comprehensive empirical work"**. In the rebuttal, we provided **more detailed experimental parameters and theoretical analysis for AMPO**. Through ablation studies, we validated the effectiveness of our RL design regarding **long-horizon consistency**. We also provided **parameter sensitivity analysis** and **additional OOD evaluations** to address potential concerns. |

We fully understand and respect ICLR's policy adjustments regarding the data incident. Thank you again for your consideration under this high workload. We hope this summary provides useful context for your recommendation.

Best regrads,

The Authors

---

### Meta-Review · Area_Chair_9Njk · 2026-01-08

**Summary:**

This submission proposes the Adaptive Social Learning (ASL) framework, which enhances language agents' adaptive reasoning ability in dynamic social interactions, addressing the lack of reasoning depth in current studies. Specifically, the framework introduces a hierarchical reasoning mode design, from intuitive responses to deep deliberations, and develops the Adaptive Mode Policy Optimization (AMPO) algorithm to enable context-aware mode switching and reasoning depth adaptation. Extensive experiments show that ASL improves task performance by 15.6% compared to GPT-4o, and AMPO outperforms GRPO by 7.0% while reducing thinking chains by 32.8%. These results highlight the efficiency and adaptability of ASL in social intelligence tasks.

**Reviewer Concerns:**

The reviewers' concerns can be summarized into the following points:

1) Reproducibility and Implementation Gaps: The paper lacks crucial details needed for exact reproduction, such as the full prompt templates, evaluator/judge prompt, hyperparameters for BC and AMPO, and the implementation specifics for the format compliance detector. These omissions hinder the ability to fully replicate the study's experiments and results.

2) Limited Theoretical Analysis: While the method is empirically validated, there is insufficient theoretical analysis regarding the properties of AMPO, such as its variance/bias behavior or the conditions under which mode-level information improves policy. A more formal or intuitive explanation would enhance confidence in the method's underlying principles.

3) Single-Turn Optimization and Long-Horizon Consistency: The paper does not adequately address the potential limitations of optimizing the model through a single-turn decomposition in multi-turn interactions, particularly regarding long-horizon strategic consistency. This issue needs either further analysis or empirical testing to assess its impact on social reasoning tasks.

**Reviewer Scores:**

Four reviewers rated 6, 8, 8, 6 respectively, and after the substantial effort during the rebuttal, one reviewers shows the potential to increase the score despite the rollback w.r.t. the openreview bug. I have checked the authors feedback about above concerns of reproduction, analysis and detailed analysis, to some extent, reviewers will be satisfied with the authors' revision, which further clarify the intuition, settings, comparison and details.

---

### Decision · Program_Chairs · 2026-01-26

Accept (Poster)